# Greedy Sampling Is Provably Efficient for RLHF

**Di Wu**
Electrical and Computer Engineering
University of Virginia
Charlottesville, VA 22903
ckk8dy@virginia.edu

**Chengshuai Shi**
Princeton Language and Intelligence
Princeton University
Princeton, NJ 08540
cs1083@princeton.edu

**Jing Yang**
Electrical and Computer Engineering
University of Virginia
Charlottesville, VA 22903
yangjing@virginia.edu

**Cong Shen**
Electrical and Computer Engineering
University of Virginia
Charlottesville, VA 22903
cong@virginia.edu

## Abstract

Reinforcement Learning from Human Feedback (RLHF) has emerged as a key technique for post-training large language models. Despite its empirical success, the theoretical understanding of RLHF is still limited, as learning the KL-regularized target with only preference feedback poses additional challenges compared with canonical RL. Existing works mostly study the reward-based Bradley-Terry (BT) preference model, and extend classical designs utilizing optimism or pessimism. This work, instead, considers the general preference model (whose practical relevance has been observed recently) and obtains performance guarantees with major, order-wise improvements over existing ones. Surprisingly, these results are derived from algorithms that directly use the empirical estimates (i.e., greedy sampling), as opposed to constructing optimistic or pessimistic estimates in previous works. This insight has a deep root in the unique structural property of the optimal policy class under the KL-regularized target, and we further specialize it to the BT model, highlighting the surprising sufficiency of greedy sampling in RLHF.

## 1   Introduction

In recent years, a growing amount of attention has been given to Reinforcement Learning from Human Feedback (RLHF), an essential component in the post-training stage of large language models (LLMs) (Bai et al., 2022; Ouyang et al., 2022; Rafailov et al., 2023; Dong et al., 2023; Zhao et al., 2023). Specifically, RLHF has been one of the key techniques powering the most widely used LLMs (Achiam et al., 2023; Bai et al., 2022; Touvron et al., 2023; Team et al., 2023). With its tremendous empirical success, researchers have started to seek a deeper theoretical understanding of RLHF (Zhu et al., 2023; Xiong et al., 2023).

Compared with canonical reinforcement learning (RL) (Sutton et al., 2018), RLHF has two key characteristics. First, rather than the absolute reward signals, the learning agent in RLHF commonly observes *preference feedback*, since during LLM training, it is easier for human annotators to compare answers than to provide absolute scores. Second, RLHF commonly considers a target with an additional penalty term as the Kullback–Leibler (KL) divergence between the learned policy and a reference policy. This target is often referred to as the *KL-regularized RL* and in the most common single-turn scenario, *KL-regularized contextual bandits (CB)*.

While these two key characteristics pose additional challenges for the analysis of RLHF, interesting theoretical guarantees have been derived. Based on Zhao et al. (2024), it has been established that learning in KL-regularized RL/CB can achieve a regret upper bound of $O(\log(T))$ for the online setting with a time horizon of $T$ (Zhao et al., 2025a), and a sample complexity of $O(\varepsilon^{-1})$ for the offline setting to obtain an $\varepsilon$-optimal policy with the single-policy coverage (Zhao et al., 2025b), both of which are instance-agnostic. These results have clear gaps to the traditional lower bounds of $\Omega(\sqrt{T})$ regret (Auer et al., 2008; Osband and Van Roy, 2016) and $\Omega(\varepsilon^{-2})$ sample complexity (Rashidinejad et al., 2021; Li et al., 2024) in canonical RL.

Despite these advances, two major limitations exist. First, the aforementioned tight performance bounds are obtained by either assuming direct reward observations (Zhao et al., 2025a,b) or restricting to the reward-based Bradley-Terry (BT) preference model (Zhao et al., 2024, 2025b). Thus, these results are not applicable to the general preference model without the concept of rewards, which has empirically demonstrated appealing flexibility and adaptability (Azar et al., 2024; Munos et al., 2023). Moreover, although the properties of KL-regularized RL/CB has been leveraged for performance analyses, they are not fully exploited to facilitate new algorithm designs. Specifically, the proposed designs still follow the classical principles of *optimism* (Auer et al., 2002; Abbasi-Yadkori et al., 2011) or *pessimism* (Rashidinejad et al., 2021; Jin et al., 2021) *in the face of uncertainty*.

In light of these two limitations, this work contributes to the theoretical understanding of RLHF by targeting the general preference model and revisiting the algorithm designs. The main results are summarized in the following seemingly counter-intuitive statement, which holds for *both online and offline settings* and *both general and BT preference models*:

*Greedy sampling with respect to the empirical estimates is provably efficient for RLHF.*

This result appears surprising, as directly leveraging the empirical estimates for sampling is typically undesirable in canonical RL/CB, both online and offline. As we will illustrate in this work, it actually has a deep root in the theoretical properties introduced by the additional KL-regularization.

Centered on the above statement, the contributions of this work can be summarized as follows:

• Under the general preference model, we demonstrate that for RLHF (in particular, KL-regularized CB), we can derive a regret upper bound of $O(\log(T))$ with a time horizon of $T$ in the online setting. Furthermore, in the offline setting, a sample complexity of $O(\varepsilon^{-1})$ can be achieved only requiring a single-policy coverage. These online and offline guarantees are the first ones of these orders for the general preference model, largely improving the previous results of orders $O(\sqrt{T})$ and $O(\varepsilon^{-2})$ obtained in Ye et al. (2024b), respectively.

• More importantly, the above results are obtained by directly using *greedy sampling* with respect to the empirical estimates. As a result, in contrast to previous works that mostly use optimistic or pessimistic estimates, there is no need for constructing upper and lower confidence bounds, which is often computationally heavy in general scenarios. This is the first time, to the best of our knowledge, that the provable efficiency of greedy sampling is established for RLHF, regardless of the preference models. Technically, this surprising result comes from the fundamental fact that, under the KL-regularized objective, every candidate optimal policy lies within a bounded likelihood ratio of the reference policy – a structural property previous studies have largely overlooked.

• To further generalize the above observation, we derive the performance bounds of greedy sampling under the BT model, which are again of orders $O(\log(T))$ and $O(\varepsilon^{-1})$ for the online and offline settings, respectively. These results match the upper bounds in previous works (Zhao et al., 2025a,b), which are derived under optimism and pessimism.

• The theoretical results (i.e., the efficiency of directly using empirical estimates) are corroborated with simulation results. In particular, under both general and BT preference models, experimental results demonstrate that the simple greedy sampling approach achieves statistically comparable performance to prior methods with more sophisticated policy constructions.

## 2 Problem Formulation and Preliminaries

Based on the standard LLM post-training pipeline of RLHF (Ouyang et al., 2022; Achiam et al., 2023), this work focuses on the following KL-regularized contextual bandits (CB) problem with

preference feedback, which is also studied in the previous literature with a theoretical focus (Zhu et al., 2023; Xiong et al., 2023; Ye et al., 2024b; Zhao et al., 2024, 2025a,b). Specifically, it is considered that a policy $\pi : \mathcal{X} \to \Delta(\mathcal{A})$ (i.e., an LLM) takes in a context $x \in \mathcal{X}$ (i.e., a prompt) and performs an action $a \in \mathcal{A}$ (i.e., a response). It is commonly assumed that the context $x$ is independently sampled from a distribution $d_0$, i.e., $x \sim d_0$. The goal of the learning agent in RLHF is to find a good policy $\pi$ that best aligns with human values.

**Preference model.** As mentioned in Section 1, to facilitate human annotators, a key characteristic of RLHF is that the learning agent receives preference feedback over actions (typically action pairs) instead of direct reward feedback. In particular, given a context $x$ and an action pair $(a^1, a^2)$, the learning agent receives a binary feedback $y \in \{0, 1\}$, with $y = 0$ meaning $a^1$ is preferred over $a^2$, denoted as $a^1 \succ a^2$ and $y = 0$ meaning $a^2$ is preferred over $a^1$, denoted as $a^2 \succ a^1$ (or $a^1 \prec a^2$).

This work first considers the following general preference model, which directly models the probability of preferring one action over the other given a context. It has attracted increasing attention recently, due to its empirical flexibility and adaptability (Azar et al., 2024; Munos et al., 2023).

**Definition 1 (General Preference Model)** *A preference model $\mathbb{P} : \mathcal{X} \times \mathcal{A} \times \mathcal{A} \to [0, 1]$ is defined such that the preference signal $y$ of any action pair $(a^1, a^2) \in \mathcal{A} \times \mathcal{A}$ under any context $x \in \mathcal{X}$ follows*

$$y \sim \mathrm{Ber}(\mathbb{P}(a^1 \succ a^2 | x, a^1, a^2)).$$

*With the definition, it is naturally observed that $\mathbb{P}(a^1 \succ a^2 | x, a^1, a^2) + \mathbb{P}(a^2 \succ a^1 | x, a^1, a^2) = 1$. To ease the presentation, it is denoted that $P^*(x, a^1, a^2) := \mathbb{P}(a^1 \succ a^2 | x, a^1, a^2)$.*

A more commonly considered preference model is the following Bradley-Terry (BT) model (Bradley and Terry, 1952), which is also studied in this work. It can be observed that the BT model is a special case of the general preference model by introducing a latent reward function.

**Definition 2 (Bradley-Terry Model)** *The Bradley-Terry model satisfies Definition 1 with the probability of $a_1$ is preferred than $a_2$ under context $x$ modeled as*

$$\mathbb{P}(a^1 \succ a^2 | x, a^1, a^2) = \frac{\exp(R^*(x, a^1))}{\exp(R^*(x, a^1)) + \exp(R^*(x, a^2))} = \sigma\big(R^*(x, a^1) - R^*(x, a^2)\big),$$

*where $R^* : \mathcal{X} \times \mathcal{A} \to [0, 1]$ is a reward function that captures the performance of one action $a \in \mathcal{A}$ under a context $x \in \mathcal{X}$ as $R^*(x, a)$.*

Because the understanding of general preference models remains preliminary and more complex than that of the BT model, the subsequent discussion focuses primarily on the former, even though our contributions address both. Also, to facilitate the presentation, we will use GP (resp., BT) to denote the general preference model (resp., the BT model).

**Learning objective.** Besides the preference model, the second key characteristic of RLHF is that a KL-regularized target is typically considered, which originates from its practical implementations (Ziegler et al., 2019; Ouyang et al., 2022; Bai et al., 2022; Rafailov et al., 2023). In terms of the general preference model, following Munos et al. (2023); Ye et al. (2024b), a game-theoretical perspective is taken where two players (referred to as the max-player and the min-player) compete in a zero-sum game with the following value for a policy pair $(\pi^1, \pi^2)$:

$$J_{\mathrm{GP}}(\pi^1, \pi^2) := \mathbb{E}_{x \sim d_0} \mathbb{E}_{a^1 \sim \pi^1, a^2 \sim \pi^2} \left[ P^*(x, a^1, a^2) - \eta^{-1} \mathrm{KL}(\pi^1, \pi_0 | x) + \eta^{-1} \mathrm{KL}(\pi^2, \pi_0 | x) \right].$$

In the above definition, $\eta \in \mathbb{R}$ is a parameter controlling the regularization and $\pi_0$ is referred to as the *reference policy*, which is typically the model checkpoint after supervised fine-tuning in LLM training. In addition, the simplified notation $\mathrm{KL}(\pi^1, \pi_0 | x) := \mathrm{KL}(\pi^1(\cdot|x) \| \pi_0(\cdot|x))$ is adopted for ease of presentation. In general, we consider the reference policy $\pi_0$ as a non-deterministic policy.

For this game, it has been demonstrated in Munos et al. (2023); Ye et al. (2024b) that there exists a unique Nash equilibrium (NE) as

$$(\pi^{1,*}, \pi^{2,*}) = (\pi_{\mathrm{GP}}^*, \pi_{\mathrm{GP}}^*) = \underset{\pi^1 \in \Pi}{\arg\max} \, \underset{\pi^2 \in \Pi}{\arg\min} \, J_{\mathrm{GP}}(\pi^1, \pi^2),$$

where $\Pi := \{\pi : \text{support}(\pi) = \text{support}(\pi_0)\}$ is the set of policies sharing the same support as $\pi_0$ so that the KL-regularization is meaningful. When there is no ambiguity, we also refer to the NE policy as the optimal policy with a slight abuse of notion.

As defined in Ye et al. (2024b), the following performance metric is defined for a learned policy $\hat{\pi}^1$ with $J_{\text{GP}}^* := J_{\text{GP}}(\pi_{\text{GP}}^*, \pi_{\text{GP}}^*)$ as

$$\Delta_{\text{GP}}(\hat{\pi}^1) := J_{\text{GP}}^* - \min_{\pi^2 \in \Pi} J_{\text{GP}}(\hat{\pi}^1, \pi^2),$$

which is always positive when $\hat{\pi}^1 \neq \pi_{\text{GP}}^*$. A policy $\hat{\pi}^1$ is said to be $\varepsilon$-*optimal* if $\Delta_{\text{GP}}(\hat{\pi}^1) \leq \varepsilon$. When measuring the performance of a sequence of policies $\{\pi_t^1 : t \in [T]\}$ (as in the online learning setting), the following definition of regret is introduced:

$$\text{Regret}_{\text{GP}}(T) := \sum\nolimits_{t \in [T]} \Delta_{\text{GP}}(\hat{\pi}_t^1).$$

With the BT model, while the above definitions can still be applied, it is more convenient to directly consider the performance in terms of rewards (Xiong et al., 2023; Zhao et al., 2024). Especially, the value of a policy $\pi$ is defined as

$$J_{\text{BT}}(\hat{\pi}) := \mathbb{E}_{x \sim d_0} \mathbb{E}_{a \sim \pi} \left[ R^*(x, a) - \eta^{-1} \text{KL}(\pi, \pi_0|x) \right].$$

The value definition naturally leads to the following performance metric of a learned policy $\hat{\pi}$ and regret of a sequence of policies $\{\hat{\pi}_t : t \in [T]\}$:

$$\Delta_{\text{BT}}(\hat{\pi}) := J_{\text{BT}}^* - J_{\text{BT}}(\hat{\pi}); \qquad \text{Regret}_{\text{BT}}(T) := \sum\nolimits_{t \in [T]} \Delta_{\text{BT}}(\hat{\pi}_t),$$

where $J_{\text{BT}}^* := J_{\text{BT}}(\pi_{\text{BT}}^*)$ with $\pi_{\text{BT}}^* := \arg\max_{\pi \in \Pi} J_{\text{BT}}(\pi)$ as the optimal policy under the BT model.

**Notation.** To ease the presentation, we also adopt the following notations: under a general preference model $P$, the expected probability of preferring an action $a$ over a distribution $\pi$ is denoted as by $P(x, a, \pi) := \mathbb{E}_{a' \sim \pi(\cdot|x)}[P(x, a, a')]$ and, similarly, the expected probability of preferring a distribution $\pi^1$ over the other one $\pi^2$ as $P(x, \pi^1, \pi^2) := \mathbb{E}_{a^1 \sim \pi^1(\cdot|x), a^2 \sim \pi^2(\cdot|x)}[P(x, a^1, a^2)]$.

## 3 Properties of the Optimal Policy Class

In this section, the key properties of the optimal policy class for the KL-regularized CB problem are discussed, which are vital to the subsequent algorithm designs and analyses. For both the general preference model and the BT model, the optimal policy class satisfies the following proposition.

**Proposition 1 (Optimal Policy Class)** *For any reward function $R^*$ and any preference model $P^*$, the corresponding optimal policy $\pi_{\text{BT}}^*$ and the corresponding NE policy $\pi_{\text{GP}}^*$ satisfy that*

$$\pi_{\text{BT}}^*, \pi_{\text{GP}}^* \in \Pi_{\mathcal{F}} := \{\pi_f : \pi_f(a|x) \propto \pi_0(a|x) \exp(\eta f(x, a)), \forall f \in \mathcal{F}\}, \tag{1}$$

*where $\mathcal{F} := \{f : f(x, a) \in [0, 1], \forall (x, a) \in \mathcal{X} \times \mathcal{A}\}$ denotes all the functions mapping context-action pairs to values in $[0, 1]$.*

A detailed proof and discussion of Proposition 1, including the two special cases (general preference model and BT model), can be found in Appendix B. Proposition 1 indicates that the learning agent only needs to consider the policies in the optimal policy class $\Pi_{\mathcal{F}}$. In addition, the following important lemma characterizing the optimal policy class can be established.

**Lemma 1** *For any $\pi_f \in \Pi_{\mathcal{F}}$, it holds that*

$$\frac{\pi_f(a|x)}{\pi_0(a|x)} \in [\exp(-\eta), \exp(\eta)], \qquad \forall (x, a) \in \mathcal{X} \times \mathcal{A}.$$

Two major implications arise from Lemma 1. First, any policy in the optimal policy class should have the same support as the reference policy, and thus be *stochastic*. Second, the candidate optimal policies and the reference policy have a *bounded ratio* between them. These two claims do not hold in canonical settings of RL or bandits (i.e., without KL-regularization): the optimal policy class is typically considered to only contain deterministic policies, and thus, the ratio between any two of them can be infinite.

# 4 Greedy Sampling for Online Learning

## 4.1 Algorithm Design

We consider a set of functions $\mathcal{P} \subset (\mathcal{X} \times \mathcal{A} \times \mathcal{A} \to \mathbb{R})$ (and set of functions $\mathcal{R} \subset (\mathcal{X} \times \mathcal{A} \to \mathbb{R})$ under the BT model) which provides a set of candidates to approximate $P^*$ (and $R^*$ for the BT model). The function $P$ in $\mathcal{P}$ satisfies $P(x, a^1, a^2) + P(x, a^2, a^1) = 1$, and we formally present the greedy sampling algorithm for online RLHF in Algorithm 1. In each iteration (Step 4), action $a_t^1$ is drawn from the learned greedy sampling policy $\hat{\pi}_t^1$ and $a_t^2$ from the reference policy $\pi_0$. After observing the preference label $y_i$, a standard maximum likelihood estimation (MLE) is performed on the currently accessible data (Step 6) in the function class $\mathcal{P}$ (and $\mathcal{R}$ under the BT model)

In Step 7, the algorithm directly updates $\hat{\pi}_{t+1}^1$ as the optimal policy with respect to the empirical estimates from MLE. This policy is referred to as **greedy sampling** because it uses the empirical estimate "as is". This is a major departure from existing online theoretically-sound RLHF designs, all of which use optimism either by adding a bonus term to the empirical estimates when designing $\hat{\pi}_t^1$ (Zhao et al., 2025a; Jin et al., 2020) or by sampling $a_t^2$ from a carefully-constructed enhancer policy $\hat{\pi}_t^2$ which maximizes the uncertainty relative to the $\hat{\pi}_t^1$ (Ye et al., 2024b; Xiong et al., 2023).

---

**Algorithm 1** Online RLHF with Greedy Sampling

1: **Input:** parameter $\eta$, reference policy $\pi_0$, GP: preference model class $\mathcal{P}$, BT: reward function class $\mathcal{R}$
2: Initialize $\hat{\pi}_t^1 \leftarrow \pi_0$
3: **for** $t = 1, \dots, T$ **do**
4:    Sample context $x_i \sim d_0$ and two actions $a_t^1 \sim \hat{\pi}_t^1(\cdot|x_i), a_t^2 \sim \pi_0(\cdot|x_i)$
5:    Observe preference label $y_i \in \{0, 1\}$
6:    Perform the maximum likelihood estimation with $\{(x_i, a_i^1, a_i^2, y_i)\}_{i=1}^t$:

GP :   $\hat{P}_t \leftarrow \underset{P \in \mathcal{P}}{\arg\max} \sum_{i \in [t]} y_i \log P(x_i, a_i^1, a_i^2) + (1 - y_i) \log P(x_i, a_i^2, a_i^1);$

BT :   $\hat{R}_t \leftarrow \underset{R \in \mathcal{R}}{\arg\max} \sum_{i \in [t]} y_i \log \sigma(R(x, a_i^1) - R(x, a_i^2)) + (1 - y_i) \log \sigma(R(x, a_i^2) - R(x, a_i^1))$

7:    Obtain $\hat{\pi}_{t+1}^1 \leftarrow \begin{cases} \text{GP :} & \text{the NE policy associated with } \hat{P}_t \\ \text{BT :} & \text{the optimal policy associated with } \hat{R}_t \end{cases}$

8: **end for**

---

## 4.2 Theoretical Analysis

The theoretical performance guarantee of greedy sampling in the online setting for RLHF is established in this section. Specifically, we derive novel regret bounds for both the general preference model and the BT model. The standard realizability assumption is first introduced, which indicates that the true preference model $P^*$ is in the considered function class $\mathcal{P}$.

**Assumption 1** *It is assumed that $\mathcal{P}$ is finite, i.e., $N_{\mathcal{P}} := |\mathcal{P}| < \infty$, and realizable, i.e., $P^* \in \mathcal{P}$.*

For clarity of presentation, we assume that $\mathcal{P}$ has finite cardinality; this assumption can be relaxed to an infinite function class using standard covering number arguments, as in Zhao et al. (2024).

Furthermore, we introduce the following concept of the Eluder dimension, which serves as a complexity measure of the considered function class.

**Definition 3 (Eluder Dimension, General Preference Model)** *Under the general preference model, for any $D_{t-1} = \{(x_i, a_i^1, a_i^2)\}_{i=1}^{t-1}$, we define the uncertainty of $(x, a^1, a^2)$ with respect to $\mathcal{P}$ as*

$$U_{\text{GP}}(\lambda, x, a^1, a^2; \mathcal{P}, \mathcal{D}_{t-1}) = \sup_{P_1, P_2 \in \mathcal{P}} \frac{|P_1(x, a^1, a^2) - P_2(x, a^1, a^2)|}{\sqrt{\lambda + \sum_{i \in [t-1]} (P_1(x, a_i^1, a_i^2) - P_2(x, a_i^1, a_i^2))^2}},$$

*and the corresponding Eluder dimension as*

$$d_{\text{GP}}(\mathcal{P}, \lambda, T) := \sup_{x_{1:T}, a_{1:T}^1, a_{1:T}^2} \sum_{t \in [T]} \min \left\{ 1, [U_{\text{GP}}(\lambda, x, a_t^1, a_t^2; \mathcal{P}, \mathcal{D}_{t-1})]^2 \right\}.$$

Especially, the uncertainty $U_{\mathrm{GP}}$ measures how different the newly sampled data $(x, a^1, a^2)$ is from the history data $\mathcal{D}_t$, and is widely adopted in the RL literature with general function approximation (Zhang, 2023; Ye et al., 2024a). The Eluder dimension captures the cumulative effect of out-of-sample data over $T$ rounds (Zhao et al., 2025a; Russo and Van Roy, 2013; Zhang, 2023).

**Theorem 1 (Online, General Preference Model)** *Under Assumption 1, for any $\delta > 0$, with probability at least $1 - \delta$, the regret of Algorithm 1 for the general preference model satisfies*

$$\mathrm{Regret}_{\mathrm{GP}}(T) = O\left(\exp(\eta) d_{\mathrm{GP}}(\mathcal{P}, \lambda, T) \log(N_{\mathcal{P}} T/\delta)\right).$$

To the best of our knowledge, Theorem 1 is the first analysis of RLHF under the general preference model that can achieve an $O(\log(T))$ regret bound. Ye et al. (2024b) has discussed the sample complexity for the general preference model in online RLHF, whose result implies an $O(\sqrt{T})$ regret bound that is significantly improved by the above result. More remarkably, Theorem 1 is achieved by using only *greedy sampling*, without the aid of optimism needed in Ye et al. (2024b).

**Proof sketch:** To ease the presentation, the expectation $x \sim d_0$ is omitted. To establish the regret bound, we first analyze the regret that occurs in each step: $J_{\mathrm{GP}}(\pi^{1,*}, \pi^{2,*}) - \min_{\pi \in \Pi} J_{\mathrm{GP}}(\hat{\pi}_t^1, \pi)$. Since a direct analysis of this per-step regret is difficult, our first novel idea is to bound it by $J_{\mathrm{GP}}(\tilde{\pi}_t^1, \tilde{\pi}_t^2) - J_{\mathrm{GP}}(\hat{\pi}_t^1, \tilde{\pi}_t^2)$ where $\tilde{\pi}_t^2$ is the best response of $\hat{\pi}_t^1$ and $\tilde{\pi}_t^1$ is the best response of $\tilde{\pi}_t^2$, both under the ground-truth preference model $P^*$. This conversion is important as it aligns the min-player's policy to be the same (i,e., $\tilde{\pi}_t^2$) in both terms. The second main novelty is to leverage a decomposition detailed in Lemma 2 that delicately makes use of the power of KL-regularization to obtain the following bound, where $\hat{\pi}_t^2 := \hat{\pi}_t^1$ and $\pi_t^f$ is an auxiliary policy in the optimal policy class:

$$J_{\mathrm{GP}}(\tilde{\pi}_t^1, \tilde{\pi}_t^2) - J_{\mathrm{GP}}(\hat{\pi}_t^1, \tilde{\pi}_t^2) \leq \eta \mathbb{E}_{a \sim \pi_t^f}\left[\left(\hat{P}_{t-1}(x, a, \hat{\pi}_t^2) - P^*(x, a, \tilde{\pi}_t^2)\right)^2\right]$$

$$\leq 2\eta \mathbb{E}_{a \sim \pi_t^f}\left[\left(P^*(x, a, \hat{\pi}_t^2) - \hat{P}_{t-1}(x, a, \hat{\pi}_t^2)\right)^2\right] + 2\eta \mathbb{E}_{a \sim \pi_t^f}\left[\left(P^*(x, a, \tilde{\pi}_t^2) - P^*(x, a, \hat{\pi}_t^2)\right)^2\right].$$

Both terms on the left-hand side can be further bounded as $O(\mathbb{E}_{a^1 \sim \pi_0, a^2 \sim \pi_0}[(b_{t-1}(x, a^1, a^2))^2])$ (ignoring the $\eta$ dependency) with a confidence interval denoted as $b_{t-1}(x, a^1, a^2)$ via a concentration result between by $P^*$ and $\hat{P}_{t-1}$ (c.f. Lemma 7), where the property stated in Lemma 1 about the optimal policy class is also leveraged to convert all intermediate policies to $\pi_0$. Aggregating the single-step regrets leads to the regret bound $\mathrm{Regret}_{\mathrm{GP}}(T) = O(d_{\mathrm{GP}}(\mathcal{P}, \lambda, T) \log(N_{\mathcal{P}} T/\delta))$. The complete proof can be found in Appendix D. ∎

Next, the analysis is extended to the Bradley-Terry model. Similarly, the corresponding realizability assumption and the Eluder dimension are introduced.

**Assumption 2** *It is assumed that $\mathcal{R}$ is finite, i.e., $N_{\mathcal{R}} := |\mathcal{R}| < \infty$, and realizable, i.e., $R^* \in \mathcal{R}$.*

**Definition 4 (Eluder Dimension, Bradley-Terry Model)** *Under the Bradley-Terry model, for any sequence $D_{t-1} = \{(x_i, a_i^1, a_i^2)\}_{i=1}^{t-1}$, we define the uncertainty of $(x, a^1, a^2)$ with respect to $\mathcal{R}$ as:*

$$U_{\mathrm{BT}}(\lambda, x, a^1, a^2; \mathcal{R}, \mathcal{D}_{t-1})$$

$$= \sup_{R_1, R_2 \in \mathcal{R}} \frac{|R_1(x, a^1) - R_1(x, a^2) - R_2(x, a^1) + R_2(x, a^2)|}{\sqrt{\lambda + \sum_{i=1}^t (R_1(x, a_i^1) - R_1(x, a_i^2) - R_2(x, a_i^1) + R_2(x, a_i^2))^2}},$$

*and the corresponding Eluder dimension as*

$$d_{\mathrm{BT}}(\mathcal{R}, \lambda, T) := \sup_{x_{1:T}, a_{1:t}^1, a_{1:T}^2} \sum_{t \in [T]} \min\{1, [U_{\mathrm{BT}}(\lambda, x, a_t^1, a_t^2; \mathcal{R}, \mathcal{D}_{t-1})]^2\}.$$

**Theorem 2 (Online, Bradley-Terry Model)** *Under Assumption 2, for any $\delta > 0$, with probability at least $1 - \delta$, the regret of Algorithm 1 for the Bradley-Terry model satisfies:*

$$\mathrm{Regret}_{\mathrm{BT}}(T) = O\left(\exp(\eta) d_{\mathrm{BT}}(\mathcal{R}, \lambda, T) \log(N_{\mathcal{R}} T/\delta)\right).$$

Theorem 2 establishes a logarithmic regret bound for greedy sampling in the Bradley-Terry model, which shares the same order as the regret achieved by using optimism in Zhao et al. (2025a). The proof of Theorem 2 largely follows that of Theorem 1 and can be found in Appendix D.2. Finally, we remark that both theorems focus on the scaling behavior with respect to $T$. In both the general model and BT model, greedy sampling achieves logarithmic regret while circumventing the need to solve the costly optimization problem that is required in optimism/pessimism-based designs, at the cost of incurring an additional multiplicative factor of $\exp(\eta)$ in the bounds from Theorems 1 and 2. A more refined analysis of other constants (such as the dependency on $\eta$) and comparison with the previous bounds can be found in the detailed proof in Appendix D.2 as well as the discussion in Appendix A.2.

## 5 Greedy Sampling for Offline Learning

### 5.1 Algorithm Design

---

**Algorithm 2** Offline RLHF with Greedy Sampling

---

1: **Input:** parameter $\eta$, reference policy $\pi_0$, pre-collected data $\mathcal{D}_0 = \{(x_i, a_i^1, a_i^2, y_i)\}_{i=1}^m$, GP: preference model class $\mathcal{P}$, BT: reward function class $\mathcal{R}$
2: Perform the maximum likelihood estimation with $\mathcal{D}_0$:

$$\texttt{GP}: \quad \hat{P} \leftarrow \arg\max_{P \in \mathcal{P}} \sum\nolimits_{i \in [m]} y_i \log P(x_i, a_i^1, a_i^2) + (1 - y_i) \log P(x_i, a_i^2, a_i^1);$$

$$\texttt{BT}: \quad \hat{R} \leftarrow \arg\max_{R \in \mathcal{R}} \sum\nolimits_{i \in [m]} y_i \log \sigma(R(x, a_i^1) - R(x, a_i^2)) + (1 - y_i) \log \sigma(R(x, a_i^2) - R(x, a_i^1))$$

3: Obtain $\hat{\pi} \leftarrow \begin{cases} \texttt{GP}: & \text{the NE policy associated with } \hat{P} \\ \texttt{BT}: & \text{the optimal policy associated with } \hat{R} \end{cases}$

---

The proposed design for the offline setting with both the general preference model and the BT model is described in Algorithm 2. Here, MLE is performed with the pre-collected offline dataset $\mathcal{D}_0$ (Step 2) and uses the empirical estimates to calculate the output policy $\hat{\pi}$ (Step 3), i.e., as a greedy sampling policy.

### 5.2 Theoretical Analysis

Although Algorithm 2 relies solely on greedy sampling using the empirical estimates, the theoretical results in this section guarantee its efficiency. We first introduce the concept of data coverage.

**Definition 5 (Data Coverage)** *Given an offline dataset $\mathcal{D}$, let $\mu^1(a|x), \mu^2(a|x)$ be the empirical distributions of the first action $a^1$ and second action $a^2$ of $\mathcal{D}$, respectively. The coverage coefficient of $\mathcal{D}$ to the reference policy $\pi^1, \pi^2$ is defined as*

$$C(\mathcal{D}, (\pi^1, \pi^2)) = \max_{a^1, a^2, x} \frac{\pi^1(a^1|x)\pi^2(a^2|x)}{\mu^1(a^1|x)\mu^2(a^2|x)}.$$

The coverage coefficient (Zhao et al., 2024; Song et al., 2024) measures how well the offline dataset $\mathcal{D}$ covers the policy pair $(\pi^1, \pi^2)$, as this pair in $C(\mathcal{D}, (\pi^1, \pi^2))$ represents the choice of $(a^1, a^2)$ in the dataset.

**Theorem 3 (Offline, General Preference Model)** *Suppose that Assumption 1 hold. With probability at least $1 - \delta$ and for $m \geq 8\eta(\exp(2\eta) + 4\eta^2 \exp(8\eta))C(\mathcal{D}_0, (\pi_0, \pi_0)) \log(N_\mathcal{P}/\delta)/\epsilon$, the output policy of Algorithm 2 is $\epsilon$-optimal.*

**Theorem 4 (Offline, Bradley-Terry Model)** *Suppose that Assumption 2 hold. With probability at least $1 - \delta$ and for $m \geq 6\eta \exp(2\eta + 1)C(\mathcal{D}_0, (\pi_0, \pi_0)) \log(N_\mathcal{R}/\delta)/\epsilon$, the output policy of Algorithm 2 is $\epsilon$-optimal.*

We note that both Theorems 3 and 4 guarantee sample complexity $O(1/\epsilon)$. To our best knowledge, Theorem 3 is the first result in offline RLHF with the general preference model that achieves sample

complexity of $O(1/\epsilon)$. For the BT model, Zhao et al. (2025b) also achieves the order of $O(1/\epsilon)$ with pessimism, and our result establishes the same order but only using greedy sampling with respect to the empirical estimates. This is an important advantage, as a greedy policy needs fewer computational resources compared to using pessimism in offline RLHF, which requires solving optimization problems *at every step* that are often computationally demanding (Zhao et al., 2025b; Ye et al., 2024b; Zhu et al., 2023).

Similarly to the online case, we present a proof sketch of Theorem 3 in the following, and the proof of Theorem 4 follows similarly. We defer the complete proofs to Appendix C.

**Proof sketch:** Similar to the proof of Theorem 1, we can bound the suboptimal gap as

$$J_{\mathrm{GP}}(\pi^{1,*}, \pi^{2,*}) - J_{\mathrm{GP}}(\hat{\pi}^1, \tilde{\pi}^2) \leq J_{\mathrm{GP}}(\tilde{\pi}^1, \tilde{\pi}^2) - J_{\mathrm{GP}}(\hat{\pi}^1, \tilde{\pi}^2)$$

$$\leq 2\eta \mathbb{E}_{\pi_{f'}} \left[ \left( P^*(x, a, \tilde{\pi}^2) - \hat{P}(x, a, \tilde{\pi}^2) \right)^2 \right] + 2\eta \mathbb{E}_{\pi_{f'}} \left[ \left( \hat{P}(x, a, \tilde{\pi}^2) - \hat{P}(x, a, \hat{\pi}^2) \right)^2 \right].$$

With this decomposition, we only need to bound the difference of two policies $\hat{\pi}_2$ and $\tilde{\pi}_2$. The second novel idea is that these two policies can be connected by $P^*$ and the MLE result $\hat{P}$. By this observation, we can bound the regret as

$$J_{\mathrm{GP}}(\pi^{1,*}, \pi^{2,*}) - J_{\mathrm{GP}}(\hat{\pi}^1, \tilde{\pi}^2) = O\left( \mathbb{E}_{x \sim d_0, a^1 \sim \mu^1, a^2 \sim \mu^2} [(\hat{P}(x, a^1, a^2) - P^*(x, a^1, a^2))^2] \right),$$

where $\mu^1$ and $\mu^2$ are the corresponding offline data distributions. The remaining analysis is relatively straightforward, focusing on bounding the deviation of the MLE estimate $\hat{P}$ from the true parameter $P^*$. This is resolved in Lemma 4, which then completes the proof. ∎

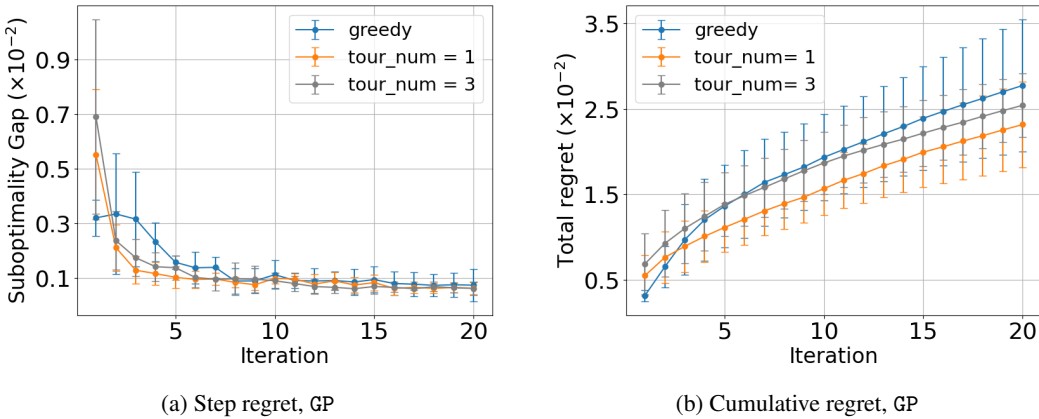

(a) Step regret, GP

(b) Cumulative regret, GP

Figure 1: The comparison between greedy sampling and optimism under the general preference model.

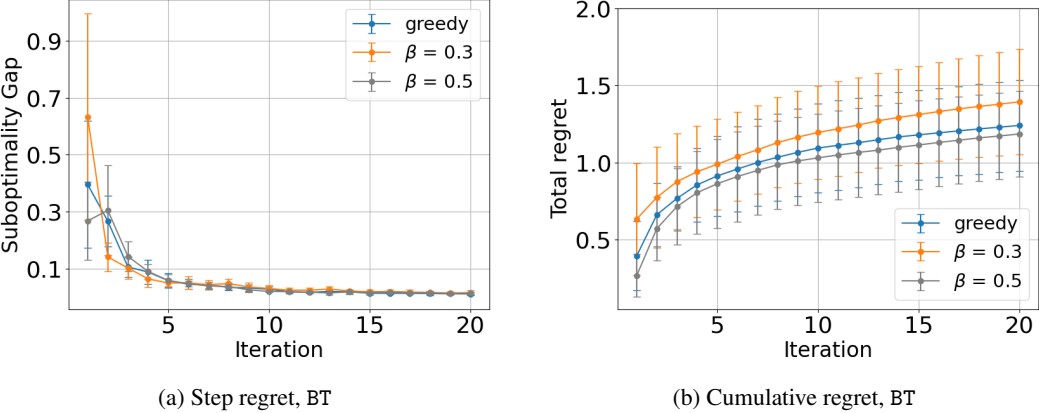

(a) Step regret, BT

(b) Cumulative regret, BT

Figure 2: The comparison between greedy sampling and optimism under the Bradley-Terry model.

# 6 Experiment

Experiments are conducted to corroborate the theoretical findings. We primarily focus on the more challenging online setting, which is also popular in practical LLM development (i.e., using iterative training as in Xiong et al. (2023); Guo et al. (2024); Xiong et al. (2024)). For both the general preference model and the BT model, we consider the linear setting with randomly sampled context vectors and 6 fixed actions. In particular, the general preference model is considered to be a linear one with dimension $k \times k \times k$ while the BT model with dimension $k \times k$, where $k$ is set to 5. Detailed experimental setups and implementation details are deferred to Appendix E.

## 6.1 Baselines

We particularly compare with previous optimism-based designs in Xiong et al. (2023); Ye et al. (2024b); Zhao et al. (2025a). For the general preference model, the original method proposed in Ye et al. (2024b) (i.e., computing an enhancer policy $\hat{\pi}_t^2$ based on uncertainty) is computationally heavy and difficult to implement. We thus adopt a tournament-style procedure that was used in Ye et al. (2024b). In particular, $a^1$ is still sampled from the greedy policy $\hat{\pi}_1^t$ as in Algorithm 1, and several candidate responses are sampled from $\hat{\pi}_1^t$ with the best of them adopted as $a^2$, which ensures a certain level of optimism. Similar rejection-based sampling as in Dong et al. (2023) is also adopted. To simulate different degrees of optimism, we use 1 and 3 as the number of responses selected in the tournament.

In the BT model with linear rewards, as considered in Xiong et al. (2023); Zhao et al. (2025a), closed-form optimism is adopted. In particular, at each step $t$, the optimistic estimate $\hat{R}_t(x, a) + \beta ||\phi(x, a) - \mathbb{E}_{x \sim d_0}[\phi(x, \pi_0)]||_{\Sigma_t^{-1}}$ is leveraged to construct $\hat{\pi}_t^1$ for sampling the first answer while the second answer is obtained from $\pi_0$ as in Algorithm 1, where $\phi(x, a)$ is the linear feature of the context-action pair and $\Sigma_t$ is the regularized Gram matrix from the collected data. To consider different levels of optimism, $\beta = 0.3$ and $0.5$ are used. It is noted that when $\beta$ equals zero, the bonus terms vanish, which returns to greedy sampling. In both settings, the regularization coefficient $\eta$ is set to 1.

## 6.2 Results

The experimental results are plotted in Figure 1 and Figure 2. We can see that both the greedy sampling and the use of optimism (tournament number equals 1 and 3 and the bonus coefficient $\beta = 0.3$ and $0.5$) have very similar convergence rate (especially w.r.t. the scaling with $T$) for the training process for both the general preference model and the BT model, which corroborates the main theoretical results in Section 4, i.e., greedy sampling is provably efficient. Taking a deeper look, we can see that the cumulative regret of greedy sampling is slightly higher than that of using optimism as iteration increases. This is aligned with our theoretical result that the greedy sampling algorithm has a slightly worse constant (see the discussion in Appendix A.2).

# 7 Related Works

**Theoretical studies of bandits and RL.** Over the past few years, a fairly comprehensive picture of canonical bandits and RL has emerged (Lattimore and Szepesvári, 2020; Agarwal et al., 2019). To obtain sublinear regrets in the online setting, one of the most widely adopted approaches is to use optimistic estimates, i.e., *optimism in the face of uncertainty* (Auer et al., 2002; Abbasi-Yadkori et al., 2011; Azar et al., 2017; Jin et al., 2018, 2020). On the other hand, in the offline setting, the principle of "pessimism" (i.e., using conservative estimates to construct policies) has become a dominant approach (Rashidinejad et al., 2021; Xie et al., 2021a,b; Jin et al., 2021; Yin and Wang, 2021; Xiong et al., 2022), which demonstrates its superiority via only requiring the single-policy coverage (i.e., offline data covering the optimal policy). Despite their statistical efficiency, one shortcoming is the required construction of confidence bounds or optimization within a confidence set, which is typically computationally infeasible, except in simple settings such as tabular and linear ones. Along this line of investigation, the theoretical studies on policy gradient methods (especially on proximal policy optimization (PPO) (Schulman et al., 2017)) are also relevant in terms of the analytical tools, such as Cai et al. (2020); Zhong and Zhang (2023); Sherman et al. (2023). However, these works still focus on learning the canonical RL target (i.e., without KL regularization) using the reward feedback.

**RLHF with the BT model.** The Bradley-Terry (BT) model is the most widely adopted preference model in RLHF and has been the basis for many empirical successes, either using logistic loss for reward model training (Ziegler et al., 2019; Stiennon et al., 2020; Ouyang et al., 2022; Bai et al., 2022) or connecting with the KL-regularized objective for direct policy optimization (Rafailov et al., 2023; Xiong et al., 2024; Rafailov et al., 2024). The majority of theoretical studies on RLHF are also focused on the BT model, with some early studies investigating the canonical target of reward-maximization (i.e., without KL regularization) (Pacchiano et al., 2021; Wang et al., 2023; Zhu et al., 2023; Zhan et al., 2023). Under the KL-regularized contextual bandits (CB) formulation (which aligns more closely with practical implementations), Xiong et al. (2023) derives the first provably efficient algorithm of RLHF, which is then followed by a line of works Xie et al. (2024); Zhong et al. (2024); Cen et al. (2024). Recently, based on the techniques proposed in Zhao et al. (2024), it is revealed that due to the KL-regularization, sharper performance guarantees can be obtained for both the online (Zhao et al., 2025a) and offline settings (Zhao et al., 2025b), as detailed in Section 1. However, these results are still based on designs that utilize optimistic or pessimistic estimates.

**RLHF with the general preference model.** While the BT model has shown strong empirical success, recent work increasingly explores more flexible and adaptable preference models that avoid implicit rewards. In particular, Azar et al. (2024) (IPO) and Munos et al. (2023) (Nash-MD) popularize the approach of leveraging the general preference model, i.e., directly model preferences among answer pairs without introducing rewards. The corresponding theoretical analysis dates back to dueling bandits (Yue and Joachims, 2009; Dudík et al., 2015), which is further extended by Wang et al. (2023) on the non-KL-regularized target. The most relevant work to our paper is Ye et al. (2024b), which studies the KL-regularized target under both the online and offline settings using the principles of optimism and pessimism. This work further improves the performance bounds in Ye et al. (2024b) and demonstrates the efficiency of directly using greedy sampling. Recent works (Zhang et al., 2024, 2025) also study the general preference model while focusing more on the performance guarantee for the planning problem instead of the learning one, as in Ye et al. (2024b) and this work.

# 8   Conclusion

This work investigated reinforcement learning from human feedback (RLHF) under the KL-regularized contextual bandits framework. Under both the general preference model and the Bradley-Terry (BT) model, it was demonstrated that the seemingly simple greedy sampling with respect to empirical estimates is provably efficient. In particular, it achieves regrets of $O(\log(T))$ in the online setting and sample complexity of $O(\varepsilon^{-1})$ with the single-policy coverage in the offline setting. These results are reported for the first time under the general preference model, and also match previous performance bounds under the BT model while eliminating the need for constructing confidence bounds (thus resulting in much lower computational overhead). The key technical insight is that KL regularization confines every candidate optimal policy within a bounded likelihood-ratio regime around the reference policy. Simulation results further corroborated the effectiveness of greedy sampling across both preference models.

# Acknowledgments

The authors thank Wei Xiong and Chenlu Ye for their helpful feedback on the initial draft of this paper. The work of Di Wu and Cong Shen was partially supported by the U.S. National Science Foundation (NSF) under grants 2143559 and 2313110, and the University of Virginia Grand Challenge Research Investments – Digital Technology Smart Infrastructure (Strategic Investment Fund Award #200). The work of Jing Yang was partially supported by the U.S. NSF under grants 2531023 and 2531789.

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

# A  Discussions

## A.1  Broad Impacts

This work focuses on the theoretical study of reinforcement learning from human feedback (RLHF) and demonstrates that it is provably efficient to use greedy sampling in both the general preference model and the Bradley-Terry (BT) model, under both the online and offline settings. While acknowledging the need for responsible usage of the proposed method, we do not foresee major negative societal impacts due to the theoretical nature of this work.

## A.2  Limitations and Future Directions

In the following, some directions worth further investigation are listed.

- From the theoretical perspective, it would be interesting to further tighten the obtained performance bounds of greedy sampling. In particular, as illustrated in the later proof, the bounds contain multiplicative constants of $\exp(\eta)$, which may be worth further studies on its necessity.
- On the empirical side, as mentioned in the main paper, the theoretical designs based on optimism or pessimism are often difficult to implement, as the bonus term or the optimization in the confidence set can be computationally demanding. With the theoretical guarantees and some empirical evidence on the effectiveness of greedy sampling in this work, more empirical, large-scale experiments would be helpful to compare greedy sampling with other methods further.
- This work focuses on the single-step setting, i.e., a contextual bandit problem, as it is the most widely adopted RLHF scenario. Still, with a growing interest in performing post-training in multi-turn scenarios (Xiong et al., 2024), it would be a promising direction to extend the theoretical results in this work to multi-step RL.

## A.3  Comparison to Related Works

Tables 1 and 2 compare our results with prior works, emphasizing the dependence on the horizon $T$ (for regret) and the sub-optimality gap $\varepsilon$ (for sample complexity). It can be observed that under the general performance model, this work substantially improves the previous results established in Ye et al. (2024b). Under the BT model, this work matches the previous performance bounds in Zhao et al. (2025a,b) while not requiring optimism or pessimism. It is noted that in the online setting, the conversion from regrets to sample complexities is performed via Lemma 14.

Table 1: Comparisons in the online setting under the general preference model and the BT model.

| Setting | Algorithm | Regret | Sample Complexity | Optimism |
|---------|-----------|--------|-------------------|----------|
| GP | Ye et al. (2024b) | – | $\widetilde{O}\big(1/\epsilon^2\big)$ | ✓ |
| | Greedy Sampling (Theorem 1) | $\widetilde{O}(\log(T))$ | $\widetilde{O}(1/\epsilon)$ | ✗ |
| BT | Xiong et al. (2023) | – | $\widetilde{O}\big(1/\epsilon^2\big)$ | ✓ |
| | Zhao et al. (2025a) | $\widetilde{O}(\log(T))$ | $\widetilde{O}(1/\epsilon)$ | ✓ |
| | Greedy Sampling (Theorem 2) | $\widetilde{O}(\log(T))$ | $\widetilde{O}\big(1/\epsilon\big)$ | ✗ |

# B  Optimal Policy Class and Value Decomposition

The optimal policy under the BT model is well-known to be a Gibbs distribution with respect to the reward function, as stated in the following proposition, whose proof can be found in Zhang (2023, Proposition 7.16) and Rafailov et al. (2023).

**Proposition 2 (Optimal Policy, BT Model)**  *For any reward function $R^*$, the corresponding optimal policy $\pi_{\mathrm{BT}}^*$ satisfies that $\pi_{\mathrm{BT}}^*(a|x) \propto \pi_0(x|a)\exp(\eta R^*(x,a))$.*

Table 2: Comparisons in the offline setting under the general preference model and the BT model.

| Setting | Algorithm | Sample Complexity | Pessimism |
|---|---|---|---|
| GP | Ye et al. (2024b) | $\widetilde{O}(1/\epsilon^2)$ | ✓ |
| | Greedy Sampling (Theorem 3) | $\widetilde{O}(1/\epsilon)$ | ✗ |
| BT | Xiong et al. (2023) | $\widetilde{O}(1/\epsilon^2)$ | ✓ |
| | Zhao et al. (2025b) | $\widetilde{O}(1/\epsilon)$ | ✓ |
| | Greedy Sampling (Theorem 4) | $\widetilde{O}(1/\epsilon)$ | ✗ |

For the general preference model, the following proposition can be established, demonstrating that the equilibrium policy still follows a similar structure.

**Proposition 3 (Equilibrium Policy, General Preference Model)** *For any preference model $P^*$, the corresponding NE policy $\pi^*_{\text{GP}}$ satisfies that $\pi^*_{\text{GP}}(a|x) \propto \pi_0(a|x) \exp(\eta P^*(x, a, \pi^*_{\text{GP}}))$.*

**Proof:** Recall that

$$(\pi^{1,*}, \pi^{2,*}) = \arg\max_{\pi^1 \in \Pi} \arg\min_{\pi^2 \in \Pi} \mathbb{E}_{x \sim d_0} P^*(x, \pi^1, \pi^2) - \eta^{-1}\text{KL}(\pi^1, \pi_0|x) + \eta^{-1}\text{KL}(\pi^2, \pi_0|x),$$

and Lemma 4 in Ye et al. (2024b) shows that $\pi^{1,*}$ is equal to $\pi^{2,*}$ and we denote them as $\pi^*_{\text{GP}}$. Thus, we have

$$\begin{aligned}
\pi^{1,*} &= \arg\max_{\pi^1 \in \Pi} \mathbb{E}_{x \sim d_0} P^*(x, \pi^1, \pi^{2,*}) - \eta^{-1}\text{KL}(\pi^1, \pi_0|x) + \eta^{-1}\text{KL}(\pi^{2,*}, \pi_0|x) \\
&= \arg\max_{\pi^1 \in \Pi} \mathbb{E}_{x \sim d_0} P^*(x, \pi_1, \pi^{2,*}) - \eta^{-1}\text{KL}(\pi_1, \pi_0|x),
\end{aligned}$$

and

$$\begin{aligned}
\pi^{2,*} &= \arg\min_{\pi^2 \in \Pi} \mathbb{E}_{x \sim d_0} P^*(x, \pi^{1,*}, \pi^2) - \eta^{-1}\text{KL}(\pi^{1,*}, \pi_0|x) + \eta^{-1}\text{KL}(\pi^2, \pi_0|x) \\
&= \arg\min_{\pi^2 \in \Pi} \mathbb{E}_{x \sim d_0} P^*(x, \pi^{1,*}, \pi^2) + \eta^{-1}\text{KL}(\pi^2, \pi_0|x) \\
&= \arg\max_{\pi^2 \in \Pi} \mathbb{E}_{x \sim d_0} [-P^*(x, \pi^{1,*}, \pi^2) - \eta^{-1}\text{KL}(\pi^2, \pi_0|x)].
\end{aligned}$$

Taking $R(x, a) = P^*(x, a, \pi^{2,*})$ and $-P^*(x, \pi^{1,*}, a)$ in Proposition 2 leads to the desired result. ∎

Proposition 1 can then be established by aggregating the above two propositions together.

To facilitate further discussions, the following notation is introduced: for the function $f(x, a) : \mathcal{X} \times \mathcal{A} \to \mathbb{R}$, it is denoted that

$$\begin{aligned}
Z_f(x) &:= \sum_{a \in \mathcal{A}} \pi_0(a|x) \exp(\eta f(x, a)) \\
\pi_f(a|x) &\propto \pi_0(a|x) \exp(\eta f(x, a)) \\
V(\pi, f) &:= \mathbb{E}_{x \sim d_0, a \sim \pi}[f(x, a) - \eta^{-1}\text{KL}(\pi(\cdot|x)||\pi_0(\cdot||x))].
\end{aligned}$$

Also, the closed-form solution to a KL-regularized objective

$$\max_{\pi \in \Pi} \mathbb{E}_{x \sim d_0, a \sim \pi}[f(x, a) - \text{KL}(\pi, \pi_0|x)]$$

is denoted as $\pi^*(a|x) \propto \pi_0(a|x) \exp(\eta f(x, a))$. Furthermore, when there is no ambiguity, we will omit $x \sim d_0$ in the expectation and use the following simplified notation:

$$\mathbb{E}_\pi[f(x, a)] := \mathbb{E}_{x \sim d_0, a \sim \pi(\cdot|x)}[f(x, a)].$$

**Lemma 2 (Value Decomposition)** *For any function $f^*(x,a), f(x,a) : \mathcal{X} \times \mathcal{A} \to \mathbb{R}$, we have*

$$V(\pi_{f^*}, f^*) - V(\pi_f, f^*) \le \eta \cdot \mathbb{E}_{\pi_{f'}}\big[\big(f(x,a) - f^*(x,a)\big)^2\big]$$

*where $f'(\cdot, \cdot) = \gamma f(\cdot, \cdot) + (1-\gamma)f^*(\cdot, \cdot)$ for some $\gamma \in [0,1]$.*

**Proof:** The following proof largely follows Lemma 3.9 in Zhao et al. (2024), which is included here for completeness. For any function $f^*(x,a), f(x,a) : \mathcal{X} \times \mathcal{A} \to \mathbb{R}$, with

$$J(f) := \log Z_f(x) - \eta \sum_{a \in \mathcal{A}} \pi_f(a|x) \cdot \Delta(x,a)$$

it can be observed that

$$V(\pi_{f^*}, f^*) - V(\pi_f, f^*)$$
$$= \mathbb{E}_{\pi_{f^*}}\left[f^*(x,a) - \frac{1}{\eta}\log\frac{\pi_{f^*}(a|x)}{\pi_0(a|x)}\right] - \mathbb{E}_{\pi_f}\left[f^*(x,a) - \frac{1}{\eta}\log\frac{\pi_f(a|x)}{\pi_0(a|x)}\right]$$
$$= \frac{1}{\eta}\mathbb{E}_{\pi_{f^*}}\left[\log\frac{\pi_0(a|x)\cdot\exp\big(\eta f^*(x,a)\big)}{\pi_{f^*}(a|x)}\right] - \frac{1}{\eta}\mathbb{E}_{\pi_f}\left[\log\frac{\pi_0(a|x)\cdot\exp\big(\eta f^*(x,a)\big)}{\pi_f(a|x)}\right]$$
$$= \frac{1}{\eta}\mathbb{E}_{x\sim d_0}\big[\log Z_{f^*}(x)\big] - \frac{1}{\eta}\mathbb{E}_{x\sim d_0}\big[\log Z_f(x)\big] - \mathbb{E}_{x\sim d_0}\left[\sum_{a\in\mathcal{A}}\pi_f(a|x)\cdot\big(f^*(x,a) - f(x,a)\big)\right]$$
$$= \frac{1}{\eta}\mathbb{E}_{x\sim d_0}[J(f^*) - J(f)]$$

where the first equality follows from the definition of the KL-divergence.

Furthermore, the first derivative of $J(f)$ with respect to $\Delta(x,a) := f(x,a) - f^*(x,a)$ can be obtained as

$$\frac{\partial J(f)}{\partial \Delta(x,a)} = \frac{\partial}{\partial \Delta(x,a)}\left[\log Z_f(x) - \eta\sum_{a'\in\mathcal{A}}\pi_f(a'|x)\cdot\Delta(x,a')\right]$$
$$= \frac{1}{Z_f(x)}\cdot\pi_0(a|x)\exp\big(\eta\cdot f(x,a)\big)\cdot\eta - \eta\cdot\pi_f(a|x)$$
$$\quad - \eta\cdot\Delta(x,a)\cdot\frac{\pi_0(a|x)\cdot\exp\big(\eta\cdot f(x,a)\big)}{Z_f(x)}\cdot\eta + \eta\cdot\Delta(x,a)\cdot\frac{\big[\pi_0(a|x)\cdot\exp\big(\eta\cdot f(x,a)\big)\big]^2}{[Z_f(x)]^2}\cdot\eta$$
$$\quad + \eta\sum_{a'\in\mathcal{A}\setminus\{a\}}\frac{\pi_0(a'|x)\cdot\exp\big(\eta\cdot f(x,a')\big)}{Z_f(x)}\cdot\eta\cdot\Delta(x,a')\cdot\frac{\pi_0(a|x)\cdot\exp\big(\eta\cdot f(x,a)\big)}{Z_f(x)}$$
$$= -\eta^2\pi_f(a|x)\Delta_f(x,a) + \eta^2\sum_{a'\in\mathcal{A}}\pi_f(a'|x)\pi_f(a|x)\Delta(x,a').$$

Then, via the mean value theorem, there exists an $f'(\cdot,\cdot) = \gamma f(\cdot,\cdot) + (1-\gamma)f^*(\cdot,\cdot)$ for some $\gamma \in [0,1]$ such that

$$\frac{1}{\eta}\mathbb{E}_{x\sim d_0}[J(f^*) - J(f)]$$
$$= \frac{1}{\eta}\mathbb{E}_{x\sim d_0}\left[\eta^2\sum_{a\in\mathcal{A}}\pi_{f'}(a|x)\cdot\gamma\cdot\big(f(x,a) - f^*(x,a)\big)^2\right]$$
$$\quad - \frac{1}{\eta}\mathbb{E}_{x\sim d_0}\left[\gamma\eta^2\sum_{a_1\in\mathcal{A}}\sum_{a_2\in\mathcal{A}}\pi_{f'}(a_1|x)\pi_{f'}(a_2|x)\big(f(x,a_1) - f^*(x,a_1)\big)\big(f(x,a_2) - f^*(x,a_2)\big)\right]$$
$$\le \eta\cdot\mathbb{E}_{\pi_{f'}}\big[\big(f(x,a) - f^*(x,a)\big)^2\big]$$

where the last inequality holds since

$$\sum_{a_1\in\mathcal{A}}\sum_{a_2\in\mathcal{A}}\pi_{f'}(a_1|x)\pi_{f'}(a_2|x)\big(f(x,a_1) - f^*(x,a_1)\big)\big(f(x,a_2) - f^*(x,a_2)\big)$$
$$= \big[\mathbb{E}_{a\sim\pi_{f'}(\cdot|x)}[f(x,a) - f^*(x,a)]\big]^2 \ge 0.$$

The proof is then concluded. ∎

# C Proofs for Offline Greedy RLHF

While we heavily focus on the online setting in the main paper, we will start with the proofs for the offline setting before those for the online setting in order to facilitate the presentation of the analyses.

## C.1 General Preference Model

We first introduce the following lemmas to bound the error of the maximum likelihood estimates.

**Lemma 3** *Given the training data $\mathcal{D} = \{(x_i, a_i^1, a_i^2, y_i)\}_{i=1}^n$, with probability $1 - \delta$ and the MLE estimator $\hat{P}$ satisfies that*

$$\sum_{i=1}^n (\hat{P}(x_i, a_i^1, a_i^2) - P^*(x_i, a_i^1, a_i^2))^2 \leq \log \frac{N_{\mathcal{P}}}{\delta}.$$

**Proof:** For any fixed function $P \in \mathcal{P}$, we first upper bound its logarithmic moment generating function as

$$\log \mathbb{E} \exp \left( \sum_{i=1}^n \log \frac{P(y_i|x_i, a_i^1, a_i^2)}{P^*(y_i|x_i, a_i^1, a_i^2)} \right)$$

$$= \log \mathbb{E} \exp \left( \sum_{i=1}^{n-1} \log \frac{P(y_i|x_i, a_i^1, a_i^2)}{P^*(y_i|x_i, a_i^1, a_i^2)} \right) + \log 2 \mathbb{E}_{y_n|x_n, a_n^1, a_n^2} \sqrt{\frac{P(y_n|x_n, a_n^1, a_n^2)}{P^*(y_n|x_n, a_n^1, a_n^2)}}$$

$$= \log \mathbb{E} \exp \left( \sum_{i=1}^{n-1} \log \frac{P(y_i|x_i, a_i^1, a_i^2)}{P^*(y_i|x_i, a_i^1, a_i^2)} \right) + \log \left( 1 - H\big(P(y_n|x_n, a_n^1, a_n^2)\|P^*(y_n|x_n, a_n^1, a_n^2)\big)^2 \right)$$

$$\leq \log \mathbb{E} \exp \left( \sum_{i=1}^{n-1} \log \frac{P(y_i|x_i, a_i^1, a_i^2)}{P^*(y_i|x_i, a_i^1, a_i^2)} \right) - H\big(P(y_n|x_n, a_n^1, a_n^2)\|P^*(y_n|x_n, a_n^1, a_n^2)\big)^2$$

$$\leq \ldots \leq - \sum_{i=1}^n H\big(P(y_i|x_i, a_i^1, a_i^2)\|P^*(y_i|x_i, a_i^1, a_i^2)\big)^2, \tag{2}$$

where $H(P\|Q)$ is the Hellinger distance defined by

$$H(P\|Q)^2 := \int_\Omega \left( \sqrt{p(z)} - \sqrt{q(z)} \right)^2 d\mu(z).$$

We continue to lower-bound the Hellinger distance by

$$\sum_{i=1}^n \left( H(P(y_i|x_i, a_i^1, a_i^2)\|P^*(y_i|x_i, a_i^1, a_i^2)) \right)^2$$

$$\geq \sum_{i=1}^n \left( \mathrm{TV}(P(y_i|x_i, a_i^1, a_i^2)\|P^*(y_i|x_i, a_i^1, a_i^2)) \right)^2$$

$$= \sum_{i=1}^n (P(x_i, a_i^1, a_i^2) - P^*(x_i, a_i^1, a_i^2))^2, \tag{3}$$

where the inequality uses the fact that for any distribution $p, q$, $H(p, q) \geq \mathrm{TV}(p, q)$ according to Theorem B.9 of Zhang (2023).

Then, by invoking Lemma 10, we obtain for any $P \in \mathcal{P}$, with probability at least $1 - \delta$,

$$\sum_{i=1}^n \log \frac{P(y_i|x_i, a_i^1, a_i^2)}{P^*(y_i|x_i, a_i^1, a_i^2)} \leq \log(N_{\mathcal{P}}/\delta) + \log \mathbb{E} \exp \left( \sum_{i=1}^n \log \frac{P(y_i|x_i, a_i^1, a_i^2)}{P^*(y_i|x_i, a_i^1, a_i^2)} \right)$$

$$\leq - \sum_{i=1}^n H\big(P(y_i|x_i, a_i^1, a_i^2)\|P^*(y_i|x_i, a_i^1, a_i^2)\big)^2 + \log(N_{\mathcal{P}}/\delta)$$

$$\leq - \sum_{i=1}^n (P(x_i, a_i^1, a_i^2) - P^*(x_i, a_i^1, a_i^2))^2 + \log(N_{\mathcal{P}}/\delta),$$

where the second inequality uses Eqn. (2), and the last inequality uses Eqn. (3). By taking $P$ as $\hat{P}$, since $\hat{P}$ is the MLE, we get

$$\sum_{i=1}^{n}(\hat{P}(x_i, a_i^1, a_i^2) - P^*(x_i, a_i^1, a_i^2))^2 \leq \sum_{i=1}^{n} \log \frac{P^*(y_i|x_i, a_i^1, a_i^2)}{\hat{P}(y_i|x_i, a_i^1, a_i^2)} + \log(N_{\mathcal{P}}/\delta)$$
$$\leq \log(N_{\mathcal{P}}/\delta),$$

which concludes the proof. ∎

**Lemma 4** *Consider two arbitrary policies $\pi_1, \pi_2$, and a set of data $\{(x_i, a_i^1, a_i^2, y_i)\}_{i=1}^{n}$ generated i.i.d. from the general preference model $P^*$ and policies $\pi_1, \pi_2$. Suppose that $\hat{P}$ is the MLE estimate, with probability at least $1 - \delta$, it holds that*

$$\frac{n}{2}\mathbb{E}_{x \sim d_0, a^1 \sim \pi^1, a^2 \sim \pi^2}[(\hat{P}(x, a^1, a^2) - P^*(x, a^1, a^2))^2] \leq 2 \log \frac{N_{\mathcal{P}}}{\delta}.$$

**Proof:** By the multiplicative Chernoff bounds (refer to Lemma 11 and Remark 2), with probability at least $1 - \delta$, for any $P \in \mathcal{P}$, we have

$$\frac{n}{2}\mathbb{E}_{x \sim d_0, a^1 \sim \pi^1, a^2 \sim \pi^2}[(P(x, a^1, a^2) - P^*(x, a^1, a^2))^2]$$
$$\leq \sum_{i=1}^{n}(P(x_i, a_i^1, a_i^2) - P^*(x_i, a_i^1, a_i^2))^2 + \log(\frac{N_{\mathcal{P}}}{\delta}).$$

Taking $P = \hat{P}$ and using Lemma 3, we can get

$$\frac{n}{2}\mathbb{E}_{x \sim d_0, a^1 \sim \pi^1, a^2 \sim \pi^2}[(\hat{P}(x, a^1, a^2) - P^*(x, a^1, a^2))^2]$$
$$\leq \sum_{i=1}^{n}(\hat{P}(x_i, a_i^1, a_i^2) - P^*(x_i, a_i^1, a_i^2))^2 + \log(\frac{N_{\mathcal{P}}}{\delta})$$
$$\leq 2 \log(\frac{N_{\mathcal{P}}}{\delta}),$$

which concludes the proof. ∎

We are now ready to prove Theorem 3.

**Proof:** [Proof of Theorem 3] The following notations are first introduced:

$$(\pi^{1,*}, \pi^{2,*}) = \arg\max_{\pi^1 \in \Pi} \arg\min_{\pi^2 \in \Pi} \mathbb{E}_{x \sim d_0} P^*(x, \pi^1, \pi^2) - \eta^{-1}\mathrm{KL}(\pi^1, \pi_0|x) + \eta^{-1}\mathrm{KL}(\pi^2, \pi_0|x),$$

$$(\hat{\pi}^1, \hat{\pi}^2) = \arg\max_{\pi^1 \in \Pi} \arg\min_{\pi^2 \in \Pi} \mathbb{E}_{x \sim d_0} \hat{P}(x, \pi^1, \pi^2) - \eta^{-1}\mathrm{KL}(\pi^1, \pi_0|x) + \eta^{-1}\mathrm{KL}(\pi^2, \pi_0|x),$$

$$\tilde{\pi}^2 = \arg\min_{\pi \in \Pi} \mathbb{E}_{x \sim d_0} P^*(x, \hat{\pi}^1, \pi) + \eta^{-1}\mathrm{KL}(\pi, \pi_0|x) = \arg\min_{\pi \in \Pi} J_{\mathrm{GP}}(\hat{\pi}^1, \pi),$$

$$\tilde{\pi}^1 = \arg\max_{\pi \in \Pi} \mathbb{E}_{x \sim d_0} P^*(x, \pi, \tilde{\pi}^2) - \eta^{-1}\mathrm{KL}(\pi, \pi_0|x) = \arg\max_{\pi \in \Pi} J_{\mathrm{GP}}(\pi, \tilde{\pi}^2),$$

and it can be noticed that the suboptimal gap can be de-composed as

$$J_{\mathrm{GP}}(\pi^{1,*}, \pi^{2,*}) - J_{\mathrm{GP}}(\hat{\pi}^1, \tilde{\pi}^2)$$
$$=[J_{\mathrm{GP}}(\pi^{1,*}, \pi^{2,*}) - J_{\mathrm{GP}}(\pi^{1,*}, \tilde{\pi}^2)] + [J_{\mathrm{GP}}(\pi^{1,*}, \tilde{\pi}^2) - J_{\mathrm{GP}}(\tilde{\pi}^1, \tilde{\pi}^2)] + [J_{\mathrm{GP}}(\tilde{\pi}^1, \tilde{\pi}^2) - J_{\mathrm{GP}}(\hat{\pi}^1, \tilde{\pi}^2)]$$
$$\leq J_{\mathrm{GP}}(\tilde{\pi}^1, \tilde{\pi}^2) - J_{\mathrm{GP}}(\hat{\pi}^1, \tilde{\pi}^2),$$

where the inequality holds as the first two terms are negative due to the above definitions.

Recall that we have

$$\pi^{1,*}(a|x) \propto \pi_0(a|x) \exp(\eta P^*(x, a, \pi^{2,*})), \quad \pi^{2,*}(a|x) \propto \pi_0(a|x) \exp(-\eta P^*(x, \pi^{1,*}, a)).$$

With Lemma 2, we can get

$$
\begin{aligned}
&J_{\text{GP}}(\tilde{\pi}^1, \tilde{\pi}^2) - J_{\text{GP}}(\hat{\pi}^1, \tilde{\pi}^2) \\
=&\mathbb{E}_{x \sim d_0} P^*(x, \tilde{\pi}^1, \tilde{\pi}^2) - \eta^{-1}\text{KL}(\tilde{\pi}^1, \pi_0|x) + \eta^{-1}\text{KL}(\tilde{\pi}^2, \pi_0|x) \\
&\quad - (\mathbb{E}_{x \sim d_0} P^*(x, \hat{\pi}^1, \tilde{\pi}^2) - \eta^{-1}\text{KL}(\hat{\pi}^1, \pi_0|x) + \eta^{-1}\text{KL}(\tilde{\pi}^2, \pi_0|x)) \\
=&\mathbb{E}_{x \sim d_0, a \sim \tilde{\pi}^1} P^*(x, a, \tilde{\pi}^2) - \eta^{-1}\text{KL}(\tilde{\pi}^1, \pi_0|x) \\
&\quad - (\mathbb{E}_{x \sim d_0, a \sim \hat{\pi}^1} P^*(x, a, \tilde{\pi}^2) - \eta^{-1}\text{KL}(\hat{\pi}^1, \pi_0|x)) \\
\leq&\eta\mathbb{E}_{x \sim d_0, a \sim \pi_{f'}} [(P^*(x, a, \tilde{\pi}^2) - \hat{P}(x, a, \hat{\pi}^2))^2] \\
\leq&2\eta\mathbb{E}_{\pi_{f'}} \left[\left(P^*(x, a, \tilde{\pi}^2) - \hat{P}(x, a, \tilde{\pi}^2)\right)^2\right] + 2\eta\mathbb{E}_{\pi_{f'}} \left[\left(\hat{P}(x, a, \tilde{\pi}^2) - \hat{P}(x, a, \hat{\pi}^2)\right)^2\right].
\end{aligned}
$$

Then, it can be bounded that

$$
\begin{aligned}
&\mathbb{E}_{\pi_{f'}} \left[\left(P^*(x, a, \tilde{\pi}^2) - \hat{P}(x, a, \tilde{\pi}^2)\right)^2\right] \\
=&\mathbb{E}_{x \sim d_0, a^1 \sim \pi_{f'}} \left[\left(\mathbb{E}_{a^2 \sim \tilde{\pi}^2}[P^*(x, a^1, a^2) - \hat{P}(x, a^1, a^2)]\right)^2\right] \\
\leq&\mathbb{E}_{x \sim d_0, a^1 \sim \pi_{f'}, a^2 \sim \tilde{\pi}^2} \left[\left(P^*(x, a^1, a^2) - \hat{P}(x, a^1, a^2)\right)^2\right] \\
\leq& \exp(2\eta)\mathbb{E}_{x \sim d_0, a^1 \sim \pi_0, a^2 \sim \pi_0} \left[\left(P^*(x, a^1, a^2) - \hat{P}(x, a^1, a^2)\right)^2\right] \\
\leq& \exp(2\eta)C(\mathcal{D}_0, (\pi_0, \pi_0))\mathbb{E}_{x \sim d_0, a^1 \sim \mu^1, a^2 \sim \mu^2} \left[\left(P^*(x, a^1, a^2) - \hat{P}(x, a^1, a^2)\right)^2\right] \\
\leq&\frac{4\exp(2\eta)C(\mathcal{D}_0, (\pi_0, \pi_0))}{n} \log(N_{\mathcal{P}}/\delta)
\end{aligned}
$$

where the first inequality is putting the square into the expectation, the second inequality is by Lemma 1, the third inequality is by the assumption of data coverage, and the final inequality is by Lemma 4.

For the other term $2\eta\mathbb{E}_{\pi_{f'}} \left[\left(\hat{P}(x, a, \tilde{\pi}^2) - \hat{P}(x, a, \hat{\pi}^2)\right)^2\right]$, we bound that by several steps. First, it is noticed that

$$
\hat{\pi}^2(a|x) \propto \pi_0(a|x) \exp(-\eta\hat{P}(x, \hat{\pi}^1, a)), \qquad \tilde{\pi}^2(a|x) \propto \pi_0(a|x) \exp(-\eta P^*(x, \hat{\pi}^1, a)),
$$

and we correspondingly define

$$
Z'(x) = \sum_a \pi_0(a|x) \exp(-\eta\hat{P}(x, \hat{\pi}^1, a)), \qquad Z''(x) = \sum_a \pi_0(a|x) \exp(-\eta P^*(x, \hat{\pi}^1, a)).
$$

It can be observed that

$$
\begin{aligned}
|Z'(x) - Z''(x)| &= |\sum_a \pi_0(a|x)[\exp(-\eta\hat{P}(x, \hat{\pi}^1, a)) - \exp(-\eta P^*(x, \hat{\pi}^1, a))]| \\
&\leq \eta|\sum_a \pi_0(a|x)(\hat{P}(x, \hat{\pi}^1, a) - P^*(x, \hat{\pi}^1, a))| \\
&= \eta|\mathbb{E}_{a^1 \sim \hat{\pi}^1, a^2 \sim \pi_0}[\hat{P}(x, a^1, a^2) - P^*(x, a^1, a^2)]| \qquad (4)
\end{aligned}
$$

where the inequality is from the mean value theorem and the fact that the bound of the derivative of $\exp(-\eta x)$ with respect to $x$ is bounded in $[-\eta, -\eta\exp(-\eta)]$ for $x \in [0, 1]$.

With the following relationship

$$
1 = \sum_a \pi_0(a|x) \geq Z'(x), Z''(x) \geq \sum_a \pi_0(a|x) \cdot \exp(-\eta) = \exp(-\eta), \qquad (5)
$$

it can be established that

$$|\hat{P}(x,a',\tilde{\pi}^2) - \hat{P}(x,a',\hat{\pi}^2)|$$

$$= |\sum_a (\tilde{\pi}^2(a|x) - \hat{\pi}^2(a|x))\hat{P}(x,a',a)|$$

$$\leq \sum_a |\tilde{\pi}^2(a|x) - \hat{\pi}^2(a|x)|$$

$$= \sum_a |\frac{\pi_0(a|x)\exp(-\eta P^*(x,\hat{\pi}^1,a))}{Z''(x)} - \frac{\pi_0(a|x)\exp(-\eta\hat{P}(x,\hat{\pi}^1,a))}{Z'(x)}|$$

$$\leq \exp(\eta)\sum_a |\pi_0(a|x)\exp(-\eta P^*(x,\hat{\pi}^1,a)) - \frac{Z''(x)\pi_0(a|x)\exp(-\eta\hat{P}(x,\hat{\pi}^1,a))}{Z'(x)}|$$

$$\leq \exp(\eta)\sum_a |\pi_0(a|x)(\exp(-\eta P^*(x,\hat{\pi}^1,a)) - \exp(-\eta\hat{P}(x,\hat{\pi}^1,a)))|$$

$$\qquad + \exp(\eta)\sum_a |\frac{Z'(x)-Z''(x)}{Z'(x)}\pi_0(a|x)\exp(-\eta\hat{P}(x,\hat{\pi}^1,a))|$$

$$\leq \eta\exp(\eta)\sum_a \pi_0(a|x)|(P^*(x,\hat{\pi}^1,a) - \hat{P}(x,\hat{\pi}^1,a))|$$

$$\qquad + \exp(\eta)\sum_a \pi_0(a|x)\frac{\eta|\mathbb{E}_{a^1\sim\hat{\pi}^1,a^2\sim\pi_0}[\hat{P}(x,a^1,a^2) - P^*(x,a^1,a^2)]|}{\exp(-\eta)}$$

$$\leq \eta\exp(\eta)\mathbb{E}_{a\sim\pi_0}|\mathbb{E}_{a'\sim\hat{\pi}^1}[\hat{P}(x,a',a) - P^*(x,a',a)]|$$

$$\qquad + \eta\exp(2\eta)|\mathbb{E}_{a^1\sim\hat{\pi}^1,a^2\sim\pi_0}[\hat{P}(x,a^1,a^2) - P^*(x,a^1,a^2)]|$$

$$\leq \eta\exp(3\eta)\mathbb{E}_{a\sim\pi_0}|\mathbb{E}_{a'\sim\pi_0}[\hat{P}(x,a',a) - P^*(x,a',a)]|$$

$$\qquad + \eta\exp(4\eta)|\mathbb{E}_{a^1\sim\pi_0,a^2\sim\pi_0}[\hat{P}(x,a^1,a^2) - P^*(x,a^1,a^2)]|$$

$$\leq 2\eta\exp(4\eta)\mathbb{E}_{a^1\sim\pi_0}\mathbb{E}_{a^2\sim\pi_0}[|\hat{P}(x,a^1,a^2) - P^*(x,a^1,a^2)|],$$

where the second inequality is from Equation (5), the fourth inequality is from Equations (5) and (4), and the last two inequalities are by Lemma 1.

Thus, we have

$$(\hat{P}(x,a',\tilde{\pi}^2) - \hat{P}(x,a,\hat{\pi}^2))^2$$
$$\leq 4\eta^2\exp(8\eta)\mathbb{E}_{a^1\sim\pi_0}\mathbb{E}_{a^2\sim\pi_0}[(\hat{P}(x,a^1,a^2) - P^*(x,a^1,a^2))^2] \qquad (6)$$
$$\leq 4\eta^2\exp(8\eta)C(D_0,(\pi_0,\pi_0))\mathbb{E}_{a^1\sim\mu^1}\mathbb{E}_{a^2\sim\mu^2}[(\hat{P}(x,a^1,a^2) - P^*(x,a^1,a^2))^2].$$

Combining the above results, we get

$$J_{\mathrm{GP}}(\pi^{1,*},\pi^{2,*}) - J_{\mathrm{GP}}(\hat{\pi}^1,\tilde{\pi}^2)$$
$$\leq 2\eta C(D_0,(\pi_0,\pi_0))(\exp(2\eta) + 4\eta^2\exp(8\eta))\mathbb{E}_{x\sim d_0,a^1\sim\mu^1,a^2\sim\mu^2}[(\hat{P}(x,a^1,a^2) - P^*(x,a^1,a^2))^2]$$
$$\leq \frac{8\eta C(D_0,(\pi_0,\pi_0))(\exp(2\eta) + 4\eta^2\exp(8\eta))}{m}\log\left(\frac{N_{\mathcal{P}}}{\delta}\right),$$

and when

$$m \geq \frac{8\eta C(D_0,(\pi_0,\pi_0))(\exp(2\eta) + 4\eta^2\exp(8\eta))}{\epsilon}\log\left(\frac{N_{\mathcal{P}}}{\delta}\right),$$

the suboptimal gap is less than $\epsilon$. ∎

## C.2 The Bradley-Terry Model

Similarly, under the BT model, we start with bounding the estimation error of the MLE.

**Lemma 5** *Given the training data $\mathcal{D} = \{(x_i, a_i^1, a_i^2, y_i)\}_{i=1}^n$, with probability $1 - \delta$ and the MLE estimator $\hat{R}$ satisfies the following estimation:*

$$\sum_{i=1}^n \left[\hat{R}(x_i, a_i^1) - \hat{R}(x_i, a_i^2) - \left(R^*(x_i, a_i^1) - R^*(x_i, a_i^2)\right)\right]^2 \leq 2e \log\left(\frac{N_\mathcal{R}}{\delta}\right).$$

**Proof:** Substituting $P(x_i, a_i^1, a_i^2)$ with $\sigma(R(x_i, a_i^1) - R(x_i, a_i^2))$ in Lemma 3, we immediately get

$$\sum_{i=1}^n \left[\sigma\left(\hat{R}(x_i, a_i^1) - \hat{R}(x_i, a_i^2)\right) - \sigma\left(R^*(x_i, a_i^1) - R^*(x_i, a_i^2)\right)\right]^2 \leq \log\left(\frac{N_\mathcal{R}}{\delta}\right).$$

With the fact that $\sigma'(r) = \sigma(r)(1 - \sigma(r)) \geq \frac{1}{2e}$ (as the reward is assumed to be bounded in $[0, 1]$), it can be established that

$$\sum_{i=1}^n \left[\hat{R}(x_i, a_i^1) - \hat{R}(x_i, a_i^2) - \left(R^*(x_i, a_i^1) - R^*(x_i, a_i^2)\right)\right]^2$$

$$\leq 2e \sum_{i=1}^n \left[\sigma(\hat{R}(x_i, a_i^1) - \hat{R}(x_i, a_i^2)) - \sigma\left(R^*(x_i, a_i^1) - R^*(x_i, a_i^2)\right)\right]^2$$

$$\leq 2e \log\left(\frac{N_\mathcal{R}}{\delta}\right),$$

which concludes the proof. ∎

**Lemma 6** *Consider arbitrary policies $\pi^1, \pi^2$, and a set of context-action pairs $\{(x_i, a_i^1, a_i^2, y_i)\}_{i=1}^n$ generated i.i.d. from the BT model where $a_i^1 \sim \pi^1, a_i^2 \sim \pi^2$. Suppose that $\hat{R}$ is the MLE estimator. We have with probability at least $1 - \delta$,*

$$\mathbb{E}_{x \sim d_0, a^1 \sim \pi^1, a^2 \sim \pi^2}\left[\left(\hat{R}(x, a^1) - \hat{R}(x, a^2) - (R^*(x, a^1) - R^*(x, a^2))\right)^2\right] \leq \frac{6e}{n} \log\left(\frac{N_\mathcal{R}}{\delta}\right).$$

**Proof:** Similar to the proof of Lemma 4, with probability at least $1 - \delta$, for any $R \in \mathcal{R}$, we have

$$\frac{n}{2} \mathbb{E}_{x \sim d_0, a^1 \sim \pi^1, a^2 \sim \pi^2}\left[\left(R(x, a^1) - R(x, a^2) - (R^*(x, a^1) - R^*(x, a^2))\right)^2\right]$$

$$\leq \sum_{i=1}^n \left(R(x_i, a_i^1) - R(x_i, a_i^2) - (R^*(x_i, a_i^1) - R^*(x_i, a_i^2))\right)^2 + \log\left(\frac{N_\mathcal{R}}{\delta}\right).$$

By taking $R = \hat{R}$ and using Lemma 5, we can get

$$\frac{n}{2} \mathbb{E}_{x \sim d_0, a^1 \sim \pi^1, a^2 \sim \pi^2}\left[\left(\hat{R}(x, a_1) - \hat{R}(x, a_2) - (R^*(x, a_1) - R^*(x, a_2))\right)^2\right]$$

$$\leq \sum_{i=1}^n \left(\hat{R}(x_i, a_i^1) - \hat{R}(x_i, a_i^2) - (R^*(x_i, a_i^1) - R^*(x_i, a_i^2))\right)^2 + \log\left(\frac{N_\mathcal{R}}{\delta}\right)$$

$$\leq (2e + 1) \log\left(\frac{N_\mathcal{R}}{\delta}\right) \leq 3e \log\left(\frac{N_\mathcal{R}}{\delta}\right),$$

which proves the lemma. ∎

**Proof:** [Proof of Theorem 4] First, the output policy $\hat{\pi}$ satisfies

$$\hat{\pi} = \arg\max_{\pi \in \Pi} \mathbb{E}_\pi[\hat{R}(x, a) - \mathrm{KL}(\pi, \pi_0|x)]$$

$$= \arg\max_{\pi \in \Pi} \mathbb{E}_\pi[\hat{R}(x, a) - b(x) - \mathrm{KL}(\pi, \pi_0|x)],$$

for any function $b(x)$, which means the policies induced by $\hat{R}(x, a)$ and $\hat{R}(x, a) - b(x)$ are the same. Thus, with

$$b(x) := \mathbb{E}_{a' \sim \pi_0(\cdot|x)}[\hat{R}(x, a') - R^*(x, a')]$$

and Lemma 2, we have

$$
\begin{aligned}
& J_{\mathrm{BT}}(\pi^*) - J_{\mathrm{BT}}(\hat{\pi}) \\
& \leq \eta \mathbb{E}_{\pi_{f'}}[(\hat{R}(x, a) - b(x) - R^*(x, a))^2] \\
& = \eta \mathbb{E}_{x \sim d_0, a^1 \sim \pi_{f'}}[(\hat{R}(x, a^1) - R^*(x, a^1) - \mathbb{E}_{a^2 \sim \pi_0}(\hat{R}(x, a^2) - R^*(x, a^2)))^2] \\
& = \eta \mathbb{E}_{x \sim d_0, a^1 \sim \pi_{f'}}[(\mathbb{E}_{a^2 \sim \pi_0}[\hat{R}(x, a^1) - R^*(x, a^1) - (\hat{R}(x, a^2) - R^*(x, a^2))])^2] \\
& = \eta \mathbb{E}_{x \sim d_0, a^1 \sim \pi_{f'}, a^2 \sim \pi_0}[(\hat{R}(x, a^1) - R^*(x, a^1) - (\hat{R}(x, a^2) - R^*(x, a^2)))^2] \\
& \leq \eta \exp(2\eta) \mathbb{E}_{x \sim d_0, a^1 \sim \pi_0, a^2 \sim \pi_0}[(\hat{R}(x, a^1) - R^*(x, a^1) - (\hat{R}(x, a^2) - R^*(x, a^2)))^2] \\
& \leq \eta \exp(2\eta) C(\mathcal{D}_0, (\pi_0, \pi_0)) \mathbb{E}_{\mu^1, \mu^2}[(\hat{R}(x, a^1) - R^*(x, a^1) - (\hat{R}(x, a^2) - R^*(x, a^2)))^2] \\
& \leq 6\eta \exp(2\eta + 1) C(\mathcal{D}_0, (\pi_0, \pi_0)) \log\left(\frac{N_{\mathcal{R}}}{\delta}\right) \frac{1}{m},
\end{aligned}
\tag{7}
$$

where the last three inequalities use Lemma 1, Lemma 6 and Definition 5. Taking

$$m \geq 6\eta \exp(2\eta + 1) C(\mathcal{D}_0, (\pi_0, \pi_0)) \log\left(\frac{N_{\mathcal{R}}}{\delta}\right) / \epsilon,$$

we have $J_{\mathrm{BT}}(\pi^*) - J_{\mathrm{BT}}(\hat{\pi}) \leq \epsilon$. ∎

## D  Proofs for Online RLHF with Greedy Sampling

### D.1  General Preference Model

Define $b_t^{\mathrm{GP}}(x, a^1, a^2) = \min\{1, \beta_{T;\mathrm{GP}} \cdot U_{\mathrm{GP}}(\lambda, x, a^1, a^2; \mathcal{P}_t, \mathcal{D}_t)\}$ and

$$\mathcal{P}_t = \left\{ P \in \mathcal{P} : \sum_{i=1}^{t} (P(x, a_i^1, a_i^2) - \hat{P}_t(x, a_i^1, a_i^2))^2 + \lambda \leq \beta_{T;\mathrm{GP}}^2 \right\},$$

where $\beta_{T;\mathrm{GP}}^2 = 2\log(N_{\mathcal{P}} T/\delta)$ and $\lambda \leq \beta_{T;\mathrm{GP}}^2/2$.

**Lemma 7** *Under Algorithm 1, we have with probability at least $1 - \delta$ for all $t \in [T]$, the uniform optimism event that $\mathcal{E}_t = \{\hat{P}_t(x, a^1, a^2) + b_t^{\mathrm{GP}}(x, a^1, a^2) - P^*(x, a^1, a^2) > 0, \forall(x, a^1, a^2) \in \mathcal{X} \times \mathcal{A} \times \mathcal{A}\}$ holds true.*

**Proof:** By Lemma 3, we have that, for all $t \in [T]$ with probability at least $1 - \delta$

$$\sum_{i=1}^{n} (\hat{P}(x_i, a_i^1, a_i^2) - P^*(x_i, a_i^1, a_i^2))^2 \leq \log\left(\frac{N_{\mathcal{P}} T}{\delta}\right) = \frac{1}{2}\beta_{T;\mathrm{GP}}^2. \tag{8}$$

Hence, we deduce that for any $(x, a^1, a^2) \in \mathcal{X} \times \mathcal{A} \times \mathcal{A}$,

$$
\begin{aligned}
|\hat{P}_t(x, a^1, a^2) - P^*(x, a^1, a^2)| & \leq \sup_{P_1, P_2 \in \mathcal{P}_t} \frac{|P_1(x, a^1, a^2) - P_2(x, a^1, a^2)|}{\sqrt{\lambda + \sum_{i=1}^{t}(P_1(x, a_i^1, a_i^2) - P_2(x, a_i^1, a_i^2))^2}} \\
& \quad \cdot \sqrt{\lambda + \sum_{i=1}^{t}(\hat{P}_t(x, a_i^1, a_i^2) - P^*(x, a_i^1, a_i^2))^2} \\
& \leq U_{\mathrm{GP}}(\lambda, x, a^1, a^2; \mathcal{P}_t, \mathcal{D}_t) \sqrt{\lambda + \frac{1}{2}\beta_{T;\mathrm{GP}}^2} \\
& \leq U_{\mathrm{GP}}(\lambda, x, a^1, a^2; \mathcal{P}_t, \mathcal{D}_t) \beta_{T;\mathrm{GP}},
\end{aligned}
$$

which concludes the proof. ∎

**Proof:** [Proof of Theorem 1] Similar to the proof in the offline setting, we define

$$(\hat{\pi}_t^1, \hat{\pi}_t^2) = \arg\max_{\pi^1 \in \Pi} \ \arg\min_{\pi^2 \in \Pi} \mathbb{E}_{x \sim d_0}[\hat{P}_{t-1}(x, \pi^1, \pi^2) - \eta^{-1}\text{KL}(\pi^1, \pi_0|x) + \eta^{-1}\text{KL}(\pi^2, \pi_0|x)],$$

$$\tilde{\pi}_t^2 = \arg\min_{\pi \in \Pi} \mathbb{E}_{x \sim d_0} P^*(x, \hat{\pi}_t^1, \pi) + \eta^{-1}\text{KL}(\pi, \pi_0|x),$$

$$\tilde{\pi}_t^1 = \arg\max_{\pi \in \Pi} \mathbb{E}_{x \sim d_0} P^*(x, \pi, \tilde{\pi}_t^2) - \eta^{-1}\text{KL}(\pi, \pi_0|x).$$

Conditioning on the event $\cup_{t \in [T]} \mathcal{E}_t$ in Lemma 7, we can bound the desired regret as follows. For each step, as shown in the proof of the offline setting, we have

$$J_{\text{GP}}(\pi^{1,*}, \pi^{2,*}) - J_{\text{GP}}(\hat{\pi}_t^1, \tilde{\pi}_t^2) \le \eta \mathbb{E}_{\pi_t^f}\left[\left(\hat{P}_{t-1}(x, a, \hat{\pi}_t^2) - P^*(x, a, \tilde{\pi}_t^2)\right)^2\right]. \tag{9}$$

By Lemma 7, we can bound Equation (9) as:

$$\mathbb{E}_{\pi_t^f}\left[\left(P^*(x, \cdot, \tilde{\pi}_t^2) - \hat{P}_{t-1}(x, \cdot, \hat{\pi}_t^2)\right)^2\right]$$

$$\le \mathbb{E}_{\pi_t^f}\left[\left(P^*(x, \cdot, \tilde{\pi}_t^2) - P^*(x, \cdot, \hat{\pi}_t^2) + P^*(x, \cdot, \hat{\pi}_t^2) - \hat{P}_{t-1}(x, \cdot, \hat{\pi}_t^2)\right)^2\right]$$

$$\le 2\mathbb{E}_{\pi_t^f}\left[\left(P^*(x, \cdot, \tilde{\pi}_t^2) - P^*(x, \cdot, \hat{\pi}_t^2)\right)^2 + \left(P^*(x, \cdot, \hat{\pi}_t^2) - \hat{P}_{t-1}(x, \cdot, \hat{\pi}_t^2)\right)^2\right]$$

$$= 2\mathbb{E}_{\pi_t^f}\left[\left(P^*(x, \cdot, \tilde{\pi}_t^2) - P^*(x, \cdot, \hat{\pi}_t^2)\right)^2\right] + 2\mathbb{E}_{\pi_t^f}\left[\left(P^*(x, \cdot, \hat{\pi}_t^2) - \hat{P}_{t-1}(x, \cdot, \hat{\pi}_t^2)\right)^2\right]$$

$$\le 2\mathbb{E}_{\pi_t^f}\left[\left(P^*(x, \cdot, \tilde{\pi}_t^2) - P^*(x, \cdot, \hat{\pi}_t^2)\right)^2\right] + 2\mathbb{E}_{\pi_t^f}\left[\left(b_{t-1}^{\text{GP}}(x, \cdot, \hat{\pi}_t^2)\right)^2\right]$$

$$\le 8\eta^2 \exp(8\eta)\mathbb{E}_{a^1 \sim \pi_0}\mathbb{E}_{a^2 \sim \pi_0}[(\hat{P}(x, a^1, a^2) - P^*(x, a^1, a^2))^2] + 2\mathbb{E}_{\pi_t^f}\left[\left(b_{t-1}^{\text{GP}}(x, \cdot, \hat{\pi}_t^2)\right)^2\right]$$

$$\le (8\eta^2 \exp(8\eta) + 2\exp(2\eta))\mathbb{E}_{a^1 \sim \pi_0}\mathbb{E}_{a^2 \sim \pi_0}\left[\left(b_{t-1}^{\text{GP}}(x, a^1, a^2)\right)^2\right])$$

$$\le (8\eta^2 \exp(9\eta) + 2\exp(3\eta))\mathbb{E}_{a^1 \sim \pi_t^1}\mathbb{E}_{a^2 \sim \pi_0}\left[\left(b_{t-1}^{\text{GP}}(x, a^1, a^2)\right)^2\right]),$$

where the bound of $\mathbb{E}_{\pi_t^f}\left[\left(P^*(x, \cdot, \tilde{\pi}_t^2) - P^*(x, \cdot, \hat{\pi}_t^2)\right)^2\right]$ follows the technical analysis in the offline setting (refer to Equation (6)) and the last two inequalities follow Lemma 1.

Thus, we have

$$\text{Regret}_{\text{GP}}(T) = \sum_{t \in [T]} J_{\text{GP}}(\pi^{1,*}, \pi^{2,*}) - J_{\text{GP}}(\hat{\pi}_t^1, \tilde{\pi}_t^2)$$

$$\le (8\eta^3 \exp(9\eta) + 2\eta \exp(3\eta)) \sum_{t \in [T]} \mathbb{E}_{x \sim d_0, a^1 \sim \pi_t^1, a^2 \sim \pi_0}[b_{t-1}^{\text{GP}}(x, a^1, a^2)^2]$$

$$\le (8\eta^3 \exp(9\eta) + 2\eta \exp(3\eta)) \sum_{t \in [T]} \mathbb{E}_{x \sim d_0, a^1 \sim \pi_t^1, a^2 \sim \pi_0}[\min\{1, U_{\text{GP}}(\lambda, x, a^1, a^2; \mathcal{P}_t, \mathcal{D}_t)\}^2]\beta_{T;\text{GP}}^2$$

$$\le 4\eta \exp(3\eta)(4\eta^2 \exp(6\eta) + 1) \log(N_{\mathcal{P}} T/\delta) d_{\text{GP}}(\mathcal{P}, \lambda, T).$$

Therefore, we have the final result

$$\text{Regret}_{\text{GP}}(T) = O(\log(N_{\mathcal{P}} T/\delta) d_{\text{GP}}(\mathcal{P}, \lambda, T)),$$

where

$$d_{\text{GP}}(\mathcal{P}, \lambda, T) := \sup_{x_{1:T}, a_{1:T}^1, a_{1:T}^2} \sum_{t=1}^{T} \min\{1, [U_{\text{GP}}(\lambda, x_t, a_t^1, a_t^2; \mathcal{P}, \mathcal{D}_{t-1})]^2\}.$$

∎

## D.2 The Bradley-Terry Model

Similar to the proof for the general preference model, we define $b_t^{\text{BT}}(x, a^1, a^2) = \min\{1, \beta_{T;\text{BT}} \cdot U_{\text{BT}}(\lambda, x, a^1, a^2; \mathcal{R}_t, \mathcal{D}_t)\}$ and

$$\mathcal{R}_t = \{R \in \mathcal{R} : \sum_{i=1}^{t} (\hat{R}_t(x_i, a_i^1) - \hat{R}_t(x_i, a_i^2) - (R^*(x_i, a_i^1) - R^*(x_i, a_i^2)))^2 + \lambda \le \beta_{T;\text{BT}}^2\},$$

where $\beta_{T;\text{BT}}^2 = 4e \log(N_{\mathcal{R}} T/\delta)$ and $\lambda \le \beta_{T;\text{BT}}^2/2$.

**Lemma 8** *Under Algorithm 1, we have with probability at least $1 - \delta$ for all $t \in [T]$, the uniform optimism event that*

$$\mathcal{E}_t = \{\hat{R}_t(x, a^1) - \hat{R}_t(x, a^2) + b_t^{\mathtt{BT}}(x, a^1, a^2) - (R^*(x, a^1) - R^*(x, a^2)) > 0,$$
$$\forall (x, a^1, a^2) \in \mathcal{X} \times \mathcal{A} \times \mathcal{A}\}$$

*holds true.*

**Proof:** By Lemma 5, for all $t \in [T]$, with probability at least $1 - \delta$,

$$\sum_{i=1}^{t} \left[\hat{R}(x_i, a_i^1) - \hat{R}(x_i, a_i^2) - \left(R^*(x_i, a_i^1) - R^*(x_i, a_i^2)\right)\right]^2 \leq 2e \log\left(\frac{N_{\mathcal{R}}T}{\delta}\right) = \frac{1}{2}\beta_{T;\mathtt{BT}}^2.$$

Hence, we deduce that for any $(x, a^1, a^2) \in \mathcal{X} \times \mathcal{A} \times \mathcal{A}$,

$$|\hat{R}_t(x, a^1) - \hat{R}_t(x, a^2) - (R^*(x, a^1) - R^*(x, a^2))|$$

$$\leq \sup_{R_1, R_2 \in \mathcal{R}} \frac{|R_1(x, a^1) - R_1(x, a^2) - R_2(x, a^1) + R_2(x, a^2)|}{\sqrt{\lambda + \sum_{i=1}^{t}(R_1(x_i, a_i^1) - R_1(x_i, a_i^2) - R_2(x_i, a_i^1) + R_2(x_i, a_i^2))^2}}$$

$$\cdot \sqrt{\lambda + \sum_{i=1}^{t}(\hat{R}_t(x_i, a_i^1) - \hat{R}_t(x_i, a_i^2) - (R^*(x_i, a_i^1) - R^*(x_i, a_i^2)))^2}$$

$$\leq U_{\mathtt{BT}}(\lambda, x, a^1, a^2; \mathcal{R}_t, \mathcal{D}_t)\sqrt{\lambda + \frac{1}{2}\beta_{T;\mathtt{BT}}^2}$$

$$\leq U_{\mathtt{BT}}(\lambda, x, a^1, a^2; \mathcal{R}_t, \mathcal{D}_t)\beta_{T;\mathtt{BT}},$$

which concludes the proof. ∎

**Proof:** [Proof of Theorem 2] Conditioning on the event $\cup_{t \in [T]}\mathcal{E}$ in Lemma 8, we can bound the desire regret as follows. For single-step regret, from the analysis for the offline setting (Equation (7)), we have

$$J_{\mathtt{BT}}(\pi^*) - J_{\mathtt{BT}}(\hat{\pi}_t)$$

$$\leq \eta \mathbb{E}_{x \sim d_0, a^1 \sim \pi_{f'}, a^2 \sim \pi_0}[(\hat{R}_{t-1}(x, a^1) - R^*(x, a^1) - (\hat{R}_{t-1}(x, a^2) - R^*(x, a^2)))^2]$$

$$\leq \eta \mathbb{E}_{x \sim d_0, a^1 \sim \pi_{f'}, a^2 \sim \pi_0}[b_{t-1}^{\mathtt{BT}}(x, a^1, a^2)^2]$$

$$\leq \eta \exp(2\eta)\mathbb{E}_{x \sim d_0, a^1 \sim \pi_t^1, a^2 \sim \pi_0}[b_{t-1}^{\mathtt{BT}}(x, a^1, a^2)^2],$$

where the last inequality is by Lemma 1. Thus

$$\mathrm{Regret}_{\mathtt{BT}}(T) = \sum_{t \in [T]} J_{\mathtt{BT}}(\pi^*) - J_{\mathtt{BT}}(\hat{\pi}_t)$$

$$\leq \eta \exp(2\eta) \sum_{t \in [T]} \mathbb{E}_{x \sim d_0, a^1 \sim \pi_t^1, a^2 \sim \pi_0}[b_{t-1}^{\mathtt{BT}}(x, a^1, a^2)^2]$$

$$\leq \eta \exp(2\eta) \sum_{t \in [T]} \mathbb{E}_{x \sim d_0, a^1 \sim \pi_t^1, a^2 \sim \pi_0}[\min\{1, U_{\mathtt{BT}}(\lambda, x, a^1, a^2; \mathcal{R}_t, \mathcal{D}_t)\}^2]\beta_{T;\mathtt{BT}}^2$$

$$\leq 4\eta \exp(2\eta + 1) \log(N_{\mathcal{R}}T/\delta) \sup_{x_{1:T}, a_{1:T}^1, a_{1:T}^2} \sum_{t=1}^{T} \min\{1, [U_{\mathtt{BT}}(\lambda, x, a^1, a^2; \mathcal{R}, \mathcal{D}_{t-1})]^2\}$$

$$\leq 4\eta \exp(2\eta + 1) \log(N_{\mathcal{R}}T/\delta) d_{\mathtt{BT}}(\mathcal{R}, \lambda, T),$$

which concludes the proof. ∎

**Remark 1** *The above theoretical results indicate that the greedy algorithms suffer from some extra constant of $\eta$ (e.g. $\exp(\eta)$) compared to using optimism. At the same time, as we have discussed in the main paper, the bonus terms in optimism are usually computationally intensive, whereas the greedy policy enjoys less computational complexity. Additionally, it is unclear whether these additional dependencies on $\eta$ are fundamental – we will explore how to tighten these bounds in future work.*

# E Experiment Details

## E.1 Implementation Considerations of the General Preference Model

**Nash equilibrium oracle approximation via an iteration method.** Given a function $P \in \mathcal{P}$, the Nash equilibrium is

$$(\pi_P^{1,*}, \pi_P^{2,*}) = (\pi_P, \pi_P) = \arg\max_{\pi^1 \in \Pi} \min_{\pi^2 \in \Pi} J_{\mathsf{GP}}(\pi^1, \pi^2). \tag{10}$$

However, Equation (10) does not have a closed form, which poses a challenge in the experiment. We note that it can be transformed to a fixed-point problem by Proposition 3:

$$\pi_P = f(\pi_P), \ f(\pi) = \frac{\pi_0(a|x) \exp(\eta \sum_{a'} \pi(a'|x) P(x, a, a'))}{\sum_a \pi_0(a|x) \exp(\eta \sum_{a'} \pi(a'|x) P(x, a, a'))}.$$

In this spirit, we use the iteration method to solve the above fixed-point problem, and the iteration has been shown to converge to the fixed-point (i.e., the Nash point) in our experiment.

**Achieve optimism by enhancer and rejection sampling.** One way to achieve optimism is by choosing $\pi^2$ as the enhancer that maximizes the uncertainty to $\pi^1$ (Ye et al., 2024b; Xiong et al., 2023). Our experiment also adopts this method to simulate the online RLHF with optimism. In the general case, $\pi^2$ does not admit a closed form. Thus, we use a tournament-style procedure to get the best response (and reject all other responses), and take the best responses at $\pi^2$ (Ye et al., 2024b; Dong et al., 2023). The number of tournament competitors can be used to control the level of optimism.

## E.2 Experiment Setup

In both the general preference model and the BT model, we limit the scope to the linear case and assume the action and context are vectors in $\mathbb{R}^k$. In the general preference model, we parameterize the preference oracle $P^* \in \mathcal{P}$ by a tensor $M$ (with size $k \times k \times k$), and in the Bradley-Terry model, we parameterize the reward function $R^* \in \mathcal{R}$ by a matrix $W$ (with size $k \times k$). We give the exact form below.

**General preference model.** We parameterize the preference oracle $P \in \mathcal{P}$ by a tensor $M$ (with size $k \times k \times k$) as

$$P : \mathcal{X} \times \mathcal{A} \times \mathcal{A} \to [0, 1], \ (x, a^1, a^2) \mapsto \frac{(a^1)^T(xM)a^2}{(a^1)^T(xM)a^2 + (a^2)^T(xM)a^1} \in [0, 1].$$

**Bradley-Terry model.** We parameterize the reward function $R \in \mathcal{R}$ by a matrix $W$ (with size $k \times k$) as

$$R : \mathcal{X} \times \mathcal{A} \to [0, 1], \ (x, a) \mapsto x^T W a.$$

For implementation, we choose $k = 5$ and first uniformly randomly sample from $[0, 1]$ to construct the ground-truth preference model parameters $M^*$ and $W^*$. We similarly sample 6 vectors from $[0, 1]^5$ as the action set $\mathcal{A}$. In each iteration, we randomly sample a vector from the uniform distribution in $[0, 1]^5$ as the context vector, and then we sample action pairs $(a^1, a^2)$ based on the policies. We run the trajectory for $T$ iterations and repeat the experiments 5 times, computing the averages and standard deviations.

## E.3 Additional Experiments

To study the influence of the regularization coefficient $\eta$, we further conduct experiments evaluating the performance of the algorithm under different $\eta$ values. The experiment setting remains the same as Section E.2. We choose $\eta = 1, 2, 3$ in the greedy sampling algorithm under both the general preference model and the BT model. The results are presented in Figure 3. We can see that under all $\eta$ values, the greedy sampling converges as our theorem has suggested.

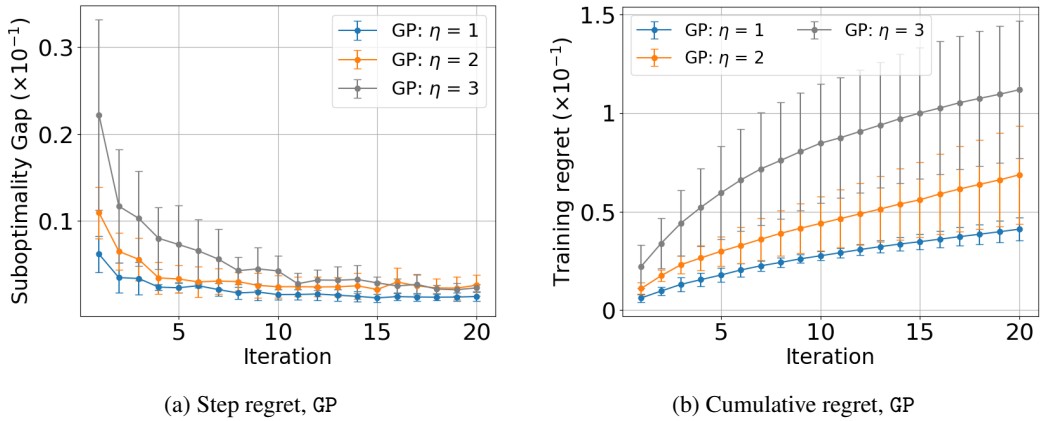

(a) Step regret, GP

(b) Cumulative regret, GP

Figure 3: The comparison of different regularization coefficients $\eta$ under the general preference model.

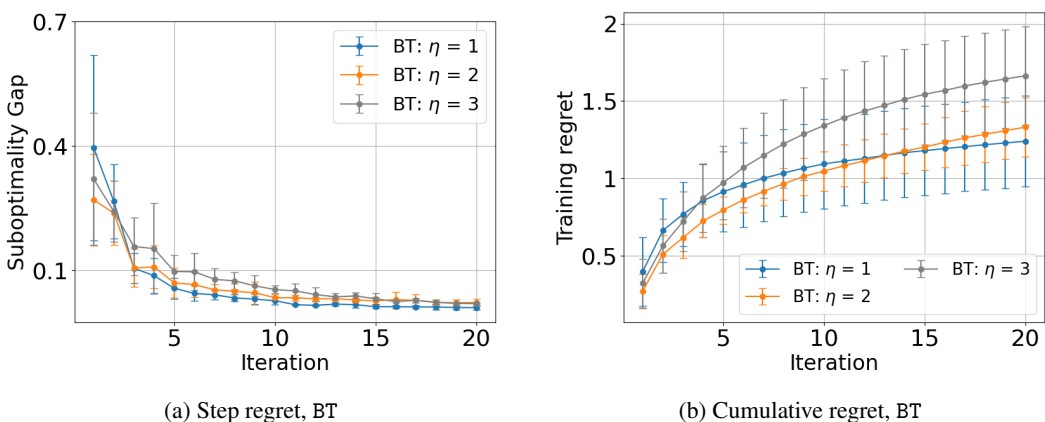

(a) Step regret, BT

(b) Cumulative regret, BT

Figure 4: Comparison of different regularization coefficients $\eta$ under the Bradley-Terry model.

# F   Auxiliary Lemmas

**Lemma 9 (Freedman's Inequality)** *Let $M, v > 0$ be fixed constants. Let $\{X_i\}_{i=1}^n$ be a stochastic process, $\{\mathcal{G}_i\}_i$ be a sequence of $\sigma$-fields, and $X_i$ be $\mathcal{G}_i$-measurable, while almost surely*

$$\mathbb{E}[X_i|\mathcal{G}_i] = 0, |X_i| \leq M, \text{ and } \sum_{i=1}^n \mathbb{E}[X_i^2|\mathcal{G}_{i-1}] \leq v.$$

*Then for any $\delta > 0$, with probability at least $1 - \delta$, it holds that*

$$\sum_{i=1}^n X_i \leq \sqrt{2v \log(1/\delta)} + \frac{2}{3} M \log(1/\delta).$$

**Lemma 10 (Martingale Exponential Inequalities)** *Consider a sequence of random functions $\xi_1(\mathcal{Z}_1), \ldots, \xi_t(\mathcal{Z}_t), \ldots$ with respect to filtration $\{\mathcal{F}_t\}$. We have for any $\delta \in (0, 1)$ and $\lambda > 0$:*

$$\mathbb{P}\left[\exists n > 0 : -\sum_{i=1}^n \xi_i \geq \frac{\log(1/\delta)}{\lambda} + \frac{1}{\lambda} \sum_{i=1}^n \log \mathbb{E}_{Z_i^{(y)}} \exp(-\lambda \xi_i)\right] \leq \delta,$$

*where $Z_t = (Z_t^{(x)}, Z_t^{(y)})$ and $\mathcal{Z}_t = (Z_1, \ldots, Z_t)$.*

**Lemma 11 (Multiplicative Chernoff Bounds)** *Assume that $X \in [0,1]$ with $\mathbb{E}X = \mu$. Then for all $\epsilon > 0$,*

$$\mathbb{P}\Big(\bar{X}_n \geq (1+\epsilon)\mu\Big) \leq \exp\left[\frac{-2n\mu\epsilon^2}{2+\epsilon}\right]$$

$$\mathbb{P}\Big(\bar{X}_n \leq (1-\epsilon)\mu\Big) \leq \exp\left[\frac{-2n\mu\epsilon^2}{2}\right].$$

*Moreover, for $t > 0$, we have*

$$\mathbb{P}\Big(\bar{X}_n \geq \mu + \sqrt{\frac{2\mu t}{n}} + \frac{t}{3n}\Big) \leq \exp(-t).$$

**Proof:** Refer to the proof of Corollary 2.18 in Zhang (2023).

**Remark 2** *The multiplicative Chernoff bounds (Lemma 11) can be expressed as follows. With probability at least $1 - \delta$:*

$$\mu \leq \bar{X}_n + \sqrt{\frac{2\mu \ln(1/\delta)}{n}}.$$

*It implies that for any $\gamma \in (0,1)$:*

$$\bar{X}_n \geq (1-\gamma)\mu - \frac{\ln(1/\delta)}{2\gamma n}.$$

∎

**Lemma 12** *Suppose $a, b \geq 0$. If $x^2 \leq a + b \cdot x$, then $x^2 \leq 2b^2 + 2a$.*

**Proof:** By solving the root of quadratic polynomial $q(x) := x^2 - b \cdot x - a$, we obtain $\max\{x_1, x_2\} = (b + \sqrt{b^2 + 4a})/2$. Hence, we have $x \leq (b + \sqrt{b^2 + 4a})/2$ provided that $q(x) \leq 0$. Then we further have

$$x^2 \leq \frac{1}{4}\Big(b + \sqrt{b^2 + 4a}\Big)^2 \leq \frac{1}{4} \cdot 2\big(b^2 + b^2 + 4a\big) \leq 2b^2 + 2a. \tag{11}$$

∎

**Lemma 13** *Let $X$ be a random variable and $0 \leq X \leq M$, we have*

$$\mathrm{Var}(X) \leq M\mathbb{E}(X).$$

**Proof:** From the Bhatia-Davis inequality, we have

$$\mathrm{Var}(X) \leq (M - \mathbb{E}(X))\mathbb{E}(X) \leq M\mathbb{E}(X).$$

∎

**Lemma 14 (Online-to-batch conversion)** *If an algorithm has a sublinear regret of $c^\dagger \cdot \log T$, then the algorithm finds an $\epsilon$-optimal policy with at most $\widetilde{\Theta}(c^\dagger/\epsilon)$ samples, where $\widetilde{\Theta}$ omits logarithmic terms of $c^\dagger/\epsilon$. Here $c^\dagger$ is a problem-dependent constant.*

**Proof:** We denote the policy sequence as $\{\pi^1, \cdots, \pi^T\}$. Then, by definition of regret, we know

$$\mathrm{Regret}(T) = TV_1^*(x_1) - \sum_{t=1}^{T} V_1^{\pi^t}(x_1)$$
$$\leq c^\dagger \log T.$$

We consider the uniform policy $\tilde{\pi} := \mathrm{Uniform}(\pi^1, \cdots, \pi^T)$. It follows that

$$V_1^*(x_1) - V_1^{\tilde{\pi}}(x_1) = V_1^*(x_1) - \frac{1}{T}\sum_{t=1}^{T} V_1^{\pi^t}(x_1) \le c^\dagger \frac{\log T}{T}.$$

It suffices to prove that

$$c^\dagger \frac{\log T}{T} \le \epsilon,$$

which is equivalent to solving

$$T \le \exp(T\epsilon/c^\dagger).$$

By using the Lambert $W$ function, we can prove that

$$T \ge \frac{W(1)c^\dagger}{\epsilon},$$

where $W(1) \ge \log(1/\epsilon) - \log\log(1/\epsilon)$. ∎

