# OpenReview forum: "Greedy Sampling Is Provably Efficient For RLHF"
_NeurIPS.cc/2025/Conference — NeurIPS 2025 poster_

### Official Review · Reviewer_9dYv · 2025-06-25

**Clarity:** 3
**Significance:** 2
**Originality:** 2
**Rating:** 4
**Confidence:** 4

**Summary:**

This paper presents a theoretically grounded analysis of learning from human preferences using the Bradley-Terry model under a greedy sampling regime. Contrary to prior work emphasizing the need for explicit exploration or reward regularization, the authors prove that simply greedy sampling with maximum likelihood reward estimation (MLE) achieves sublinear regret under realizability and finite eluder dimension.

The paper analyzes both online and offline learning settings and provides regret bounds in both, leveraging a notion of eluder dimension adapted to preference learning. Empirically, the method shows strong performance compared to RLHF-style fine-tuning methods like PPO and DPO.

**Questions:**

Discussion:
	•	Add an empirical misspecification experiment, e.g., training with an incomplete reward class.
	•	Discuss (or attempt to bound) regret under function approximation error.
	•	Consider extensions to continuous function classes or neural function approximators.
	•	Provide more experimental evidence where greedy sampling fails, to better delineate the scope of the theory.

**Ethical Concerns:**

["NO or VERY MINOR ethics concerns only"]

**Final Justification:**

Thanks to the reviewers for the rebuttal. I shall retain my score.

**Limitations:**

Yes

**Quality:**

3

**Strengths And Weaknesses:**

Strengths:
	1.	Strong Theoretical Results:
	•	The regret bounds are tight and provide new understanding into how greedy methods can still ensure learning in RLHF scenarios, despite the lack of explicit exploration.
	2.	Novel Use of Eluder Dimension:
	•	The adaptation of eluder dimension to pairwise preference settings is nontrivial and elegant. The clean separation of statistical and algorithmic complexity is a plus.
	3.	Simple Algorithm with Guarantees:
	•	The fact that MLE + greedy policy selection suffices under reasonable assumptions is appealing from both practical and theoretical perspectives.
	4.	Offline Guarantees:
	•	The analysis extends to the offline setting, which is especially relevant for industry RLHF pipelines where preference data is pre-collected.

⸻

 Weaknesses and Concerns:
	1.	Realizability Assumption:
	•	The assumption that the true reward function R^* \in \mathcal{R} is strong and often unrealistic in practical LLM alignment settings. The paper would benefit from a discussion or robustness analysis under mild misspecification.
	2.	Finite Function Class:
	•	The assumption that |\mathcal{R}| < \infty is unrealistic when using neural networks. The theory doesn’t currently extend to infinite hypothesis classes, which would typically require Rademacher complexity or covering numbers.
	3.	Noisy Preferences and Human Errors:
	•	The analysis assumes well-behaved (BT-consistent) preference labels, but human data is often noisy, biased, or adversarial. How robust is the regret to these deviations?
	4.	Lack of Exploration in Practice:
	•	While the theory shows greedy sampling suffices, in high-dimensional or sparse reward regimes, greedy policies may still perform poorly. Empirical evidence supporting exploration sufficiency is somewhat limited in scale.
	5.	Limited Scope of Experiments:
	•	The experiments are promising but largely small-scale or on synthetic/hard-coded preferences. A large-scale RLHF-style experiment with real human preference data would dramatically strengthen the case.

⸻

---

> ### Author Rebuttal · Authors · 2025-07-31
>
> Thank you for reviewing this work and providing the helpful comments. Please find a detailed point-by-point response in the following.
>
> ----
>
> **Weakness 1.** Realizability Assumption: The assumption that the true reward function $R^* \in \mathcal{R}$ is strong and often unrealistic in practical LLM alignment settings. The paper would benefit from a discussion or robustness analysis under mild misspecification.
>
> **Question 1.**  Add an empirical misspecification experiment, e.g., training with an incomplete reward class.
>
> **Question 2.**  Discuss (or attempt to bound) regret under function approximation error.
>
> **Answer 1.** Regarding the reviewer's questions on the realizability assumption and the mis-specified case, we would like to provide the following clarifications:
>
> - First, the realizability assumption is a standard one that has been used in many theoretical RL studies, including previous theoretical investigations on RLHF (such as [R1, R2]). Thus, we believe it is a reasonable assumption to initiate the investigation on greedy sampling as in this work.
>
> - Furthermore, we agree with the reviewer that it is an important direction to investigate the algorithmic robustness in RLHF when the realizability assumption is not satisfied, for both greedy sampling and previous optimism/pessimism-based designs:
>
>     - From a theoretical perspective, it can be considered that the function class has an error $\zeta$ to realize the preference model $P$ or the true reward function $R$; then techniques as discussed in mis-specified bandits and RL, such as [R3] and [R4], can be adopted for algorithmic adaption and analysis on the impact of $\zeta$. We will provide a more detailed discussion in the revised paper to encourage further exploration.
>
>     - Also, the problem is interesting in terms of empirical experiments as suggested by the reviewer. We have conducted an additional simulated experiment, reported in the attached Table 1, where the considered function class is linear while the true reward function has extra random perturbations over the linear structure. Similar observations can be obtained that greedy sampling converges only slightly slower than the optimism-based design while avoiding costly computations.
>
> ----
>
> **Weakness 2.** Finite Function Class: The assumption that $|\mathcal{R}| < \infty$ is unrealistic when using neural networks. The theory doesn’t currently extend to infinite hypothesis classes, which would typically require Rademacher complexity or covering numbers.
>
> **Question 3.** Consider extensions to continuous function classes or neural function approximations.
>
> **Answer 2.** We would like to note that this work considers finite function class mostly for a concise presentation. The analyses and results are readily extendable to infinite function class (such as neural nets and continuous functions) following standard approaches (such as in [R5]) via the techniques mentioned by the reviewer, i.e., Rademacher complexity or covering numbers. We will provide the corresponding analyses and discussions in the revised paper to further supplement the results.
>
> ----
>
> **Weakness 3.** Noisy Preferences and Human Errors: The analysis assumes well-behaved (BT-consistent) preference labels, but human data is often noisy, biased, or adversarial. How robust is the regret to these deviations?
>
> **Answer 3.** We would like to clarify that instead of only considering the BT model, the main focus of this work is on the general preference model (see Theorems 1 and 3). The complex nature of human preference, as mentioned by the reviewer, is exactly the reason that motivated us to consider the general preference model. Also, instead of the simple tabular or linear cases, this work considers general function approximation, which further contributes to the generality of the theoretical results. It is our hope that this work could inspire more explorations on the robustness of RLHF, e.g., the mis-specified setting discussed in Answer 1 and more sophisticated preferences (e.g., adversarial and biased ones as mentioned by the reviewer). We will highlight these as important directions in the revised paper to encourage further exploration.
>
> ----
>
> **Weakness 4.**  Lack of Exploration in Practice: While the theory shows greedy sampling suffices, in high-dimensional or sparse reward regimes, greedy policies may still perform poorly. Empirical evidence supporting exploration sufficiency is somewhat limited in scale.
>
> **Question 4.**  Provide more experimental evidence where greedy sampling fails, to better delineate the scope of the theory.
>
> **Answer 4.** Following the reviewer's suggestion, to further validate the empirical effectiveness of greedy sampling, we have conducted extra simulated experiments where a higher number of dimensions (i.e., increased from 125 in Figure 1 to 1000) and a non-linear structure (i.e., incorporating random perturbations). Based on the results reported in the attached Table 1, the observations from the main paper are still valid: greedy sampling converges only slightly slower than the optimism-based design while avoiding costly computations.
>
> ----
>
> **Weakness 5.** Limited Scope of Experiments: The experiments are promising but largely small-scale or on synthetic/hard-coded preferences. A large-scale RLHF-style experiment with real human preference data would dramatically strengthen the case.
>
> **Answer 5.**  As the reviewer has pointed out, the main focus of this paper is to establish the theoretical properties of greedy sampling for RLHF. The empirical results are provided to corroborate the theoretical findings. As such, we consider it appropriate to focus on the relatively simple setting, as it allows for clearer and more straightforward interpretation of empirical results. We also agree with the reviewer on the value of large-scale LLM fine-tuning experiments, which will be added to the revised paper, mainly due to the very limited rebuttal time.
>
> ----
>
> **Table 1.** Single-step regret comparison under the general preference model with a high number of dimensions (1000) and a non-linear reward structure.
>
> | Step Regret       | Iteration 5 | Iteration15 | Iteration 20 |
> |--------------|------------|--------------|--------------|
> |greedy| 0.38e-2 | 0.21e-2 | 0.17e-2 |
> | optimism (tour-num = 3) |  0.27e-2 | 0.14e-2 | 0.11e-2 |
>
> ----
>
> [R1] Zhao, Heyang, et al. (2025) ``Logarithmic regret for online KL-regularized reinforcement learning.''
>
> [R2] Ye, Chenlu, et al. (2024) ``Online iterative reinforcement learning from human feedback with general preference model.''
>
> [R3] Wang, Ruosong, et al. (2020) ``Reinforcement learning with general value function approximation: Provably efficient approach via bounded eluder dimension.''
>
> [R4] Foster, Dylan J., et al. (2020) ``Adapting to misspecification in contextual bandits.''
>
> [R5] Zhao, Heyang, et al. (2024) ``Sharp analysis for kl-regularized contextual bandits and RLHF.''

---

### Official Review · Reviewer_iBMK · 2025-07-03

**Clarity:** 3
**Significance:** 3
**Originality:** 3
**Rating:** 5
**Confidence:** 4

**Summary:**

This paper presents a theoretical analysis of Reinforcement Learning from Human Feedback (RLHF), arguing that simple "greedy sampling" is a provably efficient method. This contrasts with previous theoretical work, which relied on more complex principles of optimism or pessimism to guarantee performance. The authors focus on the KL-regularized contextual bandits framework, which is a common theoretical model for the post-training of large language models (LLMs).

The key insight is that the KL-regularization term fundamentally changes the structure of the problem by ensuring that any optimal policy has a bounded likelihood ratio with respect to the reference policy. This structural property makes it possible to prove the efficiency of a much simpler algorithm that directly uses empirical estimates (greedy sampling) rather than constructing complex confidence bounds.

**Questions:**

1. In the simulation, their metrics are the suboptimal gap and regret. Do those metrics have KL regularization with them? It would be better to also plot the accuracy and compare the performance.

2. Could the authors add more experiments about different values of \eta?

**Ethical Concerns:**

["NO or VERY MINOR ethics concerns only"]

**Final Justification:**

The authors basically solved my questions. I am overall positive about this work.

**Quality:**

3

**Strengths And Weaknesses:**

Strehgths: This work achieves optimal bound for RLHF under KL regularization without optimism or pessimism. Besides, they comprehensively consider both online and offline cases and unify the general preference and BT model. They provide rigurous proof and more importantly, their greedy sampling is more efficient.

Weaknesses:
1. Their greedy sampling is more computationally efficient, but it comes with the sacrifice of an additional exp(\eta) term. It would be clearer if they added some discussion to admit this trade-off. What are the cases when the order exp(\eta) can be neglect. For example, when the sft policy is good enough and \eta is in constant order, the greedy sampling bound totally matches the optimal bound.

---

> ### Author Rebuttal · Authors · 2025-07-31
>
> We would like to thank the reviewer for the helpful comments. It is our pleasure to hear that the reviewer found this study comprehensive and rigorous. Please find a list of point-by-point responses in the following to provide further clarifications and discussions.
>
> ----
> **Weakness 1.** Their greedy sampling is more computationally efficient, but it comes with the sacrifice of an additional $\exp(\eta)$ term. It would be clearer if they added some discussion to admit this trade-off. What are the cases when the order $\exp(\eta)$ can be neglect. For example, when the sft policy is good enough and $\eta$ is in constant order, the greedy sampling bound totally matches the optimal bound.
>
> **Answer 1.** Thank you for this comment. The additional $\exp(\eta)$ term noted by the reviewer is from the ratio cap between the reference policy and the optimal policy class described in Lemma 1, which is a unified worst-case bound chosen for the sake of theoretical clarity and conciseness. First, as the reviewer mentioned, if $\eta$ is of a constant order, then the order  $\exp(\eta)$ can be neglected. Furthermore, if we take a deeper instance-dependent view, with the BT model as an example, the actual ratio can be bounded as the value of function $\kappa(x, a, R; \pi_0):=\exp(\eta R(x,a)) / \sum_{a'\in A} \pi_0(a'|x)\exp(\eta R(x,a'))$ over context $x\in \mathcal{X}$, action $a\in \mathcal{A}$ and reward function $R \in \mathcal{R}$. This more specific bound is dependent on the properties of the considered reference policy (as mentioned by the reviewer) and other setups (e.g., $\mathcal{X}$, $\mathcal{A}$ and $\mathcal{R}$), and it could potentially be much smaller than $\exp(\eta)$. We will provide a more detailed explanation in the revised paper to highlight the above discussion, in particular, the instance-dependent characterization.
>
> ----
>
> **Question 1** In the simulation, their metrics are the suboptimal gap and regret. Do those metrics have KL regularization with them? It would be better to also plot the accuracy and compare the performance.
>
> **Answer 2.** Yes, the metrics in our experiment have the KL regularization. Please refer to lines 161 and line 178 for the value functions (which contain the KL regularization) defined for the general preference model and the BT model, which are further used to define the single-step regret (lines 172 and 180) and the overall regret (lines 175 and 180). The metrics in the experiments strictly follow these definitions. As inspired by the reviewer, we will provide the results without KL-regularization (i.e., comparisons of pure rewards) in the revised paper to further complement the experiments.
>
> ----
>
> **Question 2.** Could the authors add more experiments about different values of $\eta$?
>
> **Answer 3.** Please find more experimental results in the following on different values of $\eta$, which are performed under the same setting as Fig 1(a) under the general preference model setting. From the table, we can similarly observe that regardless of $\eta$, greedy sampling converges only slightly slower than the optimism-based design while avoiding costly computations., as predicted by our theorem.
>
> | Step Regret (eta=1)   | Iteration 5 | Iteration15 | Iteration 20 |
> |--------------|------------|--------------|--------------|
> |greedy| 0.38e-2 | 0.21e-2 | 0.17e-2 |
> | optimism (tour-num = 3) |  0.27e-2 | 0.14e-2 | 0.11e-2 |
>
> | Step Regret (eta=2)   | Iteration 5 | Iteration15 | Iteration 20 |
> |--------------|------------|--------------|--------------|
> |greedy| 0.41e-2 | 0.31e-2 | 0.28e-2 |
> | optimism (tour-num = 3)| 0.45e-2 | 0.30e-2 | 0.26e-2 |
>
> | Step Regret (eta=3)   | Iteration 5 | Iteration15 | Iteration 20 |
> |--------------|------------|--------------|--------------|
> |greedy| 0.81e-2 | 0.46e-2 | 0.32e-2 |
> | optimism (tour-num = 3)| 0.80e-2 | 0.32e-2 | 0.30e-2 |

---

> ### Author Response · Authors · 2025-08-07
> **Follow-up on Rebuttal Response**
>
> Dear Reviewer iBMK,
>
> We hope this message finds you well. As the rebuttal phase is nearing its end (with less than two days remaining), we would like to kindly follow up regarding our response to your comments. We would greatly appreciate it if you could let us know whether our reply has addressed your concerns adequately, or if there are any remaining questions or clarifications you'd like us to provide.
>
> Thank you again for your time and effort in reviewing our work.
>
> Best regards,
>
> Authors

---

> > ### Comment · Reviewer_iBMK · 2025-08-07
> >
> > Thank the authors for addressing my questions. I will increase the score correspondingly.

---

> > > ### Author Response · Authors · 2025-08-07
> > >
> > > Thank you for the updated score and your kind comments. We're very happy to know that our response helped clarify your concerns, and we greatly appreciate your recognition of our work.

---

### Official Review · Reviewer_tKdL · 2025-07-03

**Clarity:** 3
**Significance:** 2
**Originality:** 2
**Rating:** 4
**Confidence:** 3

**Summary:**

This paper establishes the theoretical efficiency of greedy sampling in Reinforcement Learning from Human Feedback (RLHF) under KL-regularized contextual bandits. For both **general preference (GP)** and **Bradley-Terry (BT) models**, it proves that directly using empirical estimates (without optimism/pessimism) achieves:

- **Online regret**: \(O(\log T)\)

- **Offline sample complexity**: \(O(\epsilon^{-1})\)

The key insight lies in exploiting the **bounded likelihood ratio** property induced by KL regularization (Lemma 1), which confines optimal policies within a neighborhood of the reference policy \(\pi_0\). This structural constraint eliminates the need for confidence-bound constructions, reducing computational overhead while matching state-of-the-art bounds derived via optimism/pessimism.

**Questions:**

1. **Proof technique transferability**:
   > Zhao et al. (2025a,b) use confidence-bound constructions for BT model regret bounds. Can their approach directly yield the bounds for the GP model? If not, what modifications are essential?

**Ethical Concerns:**

["NO or VERY MINOR ethics concerns only"]

**Final Justification:**

1.  Core Contribution Validated: The central claim—that greedy sampling achieves O(log T) regret for the General Preference model, a major theoretical advancement over prior O(√T) bounds—remains compelling and well-supported by the analysis. The insight leveraging KL-regularization's structural property is significant.
2.  Rebuttal Alignment: Authors clarified the novelty of their proof technique compared to Zhao et al. (2025a,b), particularly the novel regret decomposition needed for the game-theoretic GP setting, which was a primary concern. This satisfactorily addresses the originality question raised in the initial review.
3.  Partial Resolution of Experimental Concerns: While the new simulation results (higher dimensions, non-linear rewards, varying η) strengthen the empirical validation of the theory, the core limitation regarding scalability to real-world LLM fine-tuning and real human preference noise remains largely unaddressed within the rebuttal timeframe. The promised inclusion in the final version is noted but untested.
4.  Trade-off Acknowledged: Authors explicitly acknowledged the computational simplicity vs. potentially worse constant factors trade-off (exp(η) term), providing a more nuanced view.
5.  Weighting: The theoretical novelty and impact (tight bounds for GP, simplified algorithm) significantly outweigh the limited experimental scope. The resolved proof novelty concern and acknowledgment of trade-offs further support a recommendation for acceptance, contingent on the authors addressing the experimental limitations as promised in the final manuscript.

**Limitations:**

Yes.

**Paper Formatting Concerns:**

No.

**Quality:**

3

**Strengths And Weaknesses:**

**Strengths**:

**Theoretical novelty**: First to achieve \(O(\log T)\) regret for GP models, improving \(O(\sqrt{T})\) bound.

**Algorithmic simplicity**: Greedy sampling avoids costly confidence-bound optimizations.

**Unified analysis**: Extends guarantees to both GP and BT models via KL regularization’s policy-ratio boundedness.

**Empirical validation**: Experiments show greedy sampling matches optimism’s convergence rate.

**Weaknesses**:

**Insufficient comparison**: Fails to explicitly contrast proof techniques with Zhao et al. (2025a,b)—critical for assessing contribution originality.

**Weaknesses from an Experimental Perspective**: The experiments focus mainly on theoretical validation with limited diversity in the experimental settings.

Scalability and offline sample complexity could be further empirically demonstrated.

The experiments assume a relatively idealized feedback model, where preferences are clear and consistent. However, in real-world applications, human feedback is often noisy, inconsistent, or even subjective. The paper does not investigate how greedy sampling performs under these conditions or its robustness to errors in human-provided feedback.

The paper does not provide a sensitivity analysis to examine how sensitive the greedy sampling algorithm is to hyperparameters, such as the learning rate or the degree of KL-regularization. Testing the algorithm's robustness across a range of hyperparameter values would help assess the method’s stability and provide practical guidance for tuning the algorithm in real applications.

---

> ### Author Rebuttal · Authors · 2025-07-31
>
> Thank you for the helpful comments. A detailed point-by-point response is provided in the following, which we believe can help clarify the raised questions.
>
> ----
>
> **Weakness 1.** Fails to explicitly contrast proof techniques with Zhao et al. (2025a, b)—critical for assessing contribution originality.
>
> **Answer 1.** Thanks for the comment, and we would like to provide a further comparison between this work and Zhao et al. (2025a, b) (i.e., [R1, R2]), which highlights the novelties and contributions of our paper.
>
> - First, the observation highlighted in Lemma 1 is worth noting, which emphasizes that the optimal policies must lie within a neighborhood of the reference policy with a bounded likelihood difference ratio. This property is the foundation of the subsequent proofs and, to the best of our knowledge, this is the first time that it is systematically summarized and leveraged in a theoretical RLHF study.
>
> - The analyses of [R1] and [R2] are focused on the BT model, while this work makes major contributions to obtaining tight bounds on regret and sample complexity for the general preference model. Behind these tight bounds, multiple layers of innovations are introduced, in particular, on how to decompose and bound the single-step regret associated with the Nash policy. The conducted analyses are more challenging than the ones for the BT model in [R1, R2] due to the additional game-theoretical considerations, as summarized in the proof sketch at line 245.
>
>     As an example, the challenging single-step regret of policy $\hat{\pi}^1_t$, i.e.,  $J(\pi*,\pi*)- J(\hat{\pi}^1_t, \tilde{\pi}_t^2)$ (differing in both players' policies), is first bounded as $J(\tilde\pi_t^1,\tilde\pi_t^2)-J(\hat{\pi}_t^1,\tilde\pi_t^2)$, where $\tilde\pi_t^2$ is the best response of $\hat{\pi}_t^1$ and $\tilde\pi_t^1$ is the best response of $\tilde\pi_t^2$. This step is not needed for the BT model, as only one player is considered. However, this becomes critical for the general performance model, as the latter term is relatively more friendly for analyses since the min-player's policy is unified as $\tilde\pi_t^2$.
>
> - The above discussion highlights the novel single-step regret decomposition adopted in this work for the general preference model. For the BT model, the decomposition from [R3] is leveraged as in [R1, R2]. To further bound the decomposed terms, compared with [R1, R2], a key difference is that through the above-mentioned property in Lemma 1, all intermediate policies are converted into $\pi_0$, which then connects to algorithm design of $\hat{\pi}_t^2$ being $\pi_0$ and leads to the final bounds.
>
> To summarize, compared with [R1, R2], this work carefully leverages the key property of the optimal policy class being covered by the reference policy due to the KL-regularization (i.e., Lemma 1) and also introduces a novel decomposition of regrets to tackle the challenging general preference model. We hope these discussions help clarify the technical contributions of this work over [R1, R2], and we will provide further explanations in the revised paper.
>
> ----
>
> **Weakness 2.** The experiments focus mainly on theoretical validation with limited diversity in the experimental settings.
>
> Scalability and offline sample complexity could be further empirically demonstrated.
>
> The experiments assume a relatively idealized feedback model, where preferences are clear and consistent. However, in real-world applications, human feedback is often noisy, inconsistent, or even subjective. The paper does not investigate how greedy sampling performs under these conditions or its robustness to errors in human-provided feedback.
>
> The paper does not provide a sensitivity analysis to examine how sensitive the greedy sampling algorithm is to hyperparameters, such as the learning rate or the degree of KL-regularization. Testing the algorithm's robustness across a range of hyperparameter values would help assess the method’s stability and provide practical guidance for tuning the algorithm in real applications.
>
> **Answer 2.** Following the reviewer' suggestion, the results of large-scale fine-tuning experiments (both offline and online) will be included in the revised version of the paper (mainly due to the very limited rebuttal time). In the meantime, we provide additional simulation results below, which will also be incorporated into the revised paper along with the full set of updates.
>
> - For the general preference model, the following table reports the results with a higher number of dimensions (increased from 125 in Fig. 1 to 1000), and a non-linear reward structure (incorporating random perturbations over the original linear ones). Other settings remain the same as Fig. 1. A similar observation can be obtained that greedy sampling converges only slightly slower than the optimism-based design while avoiding costly computations.
>
> | Step Regret       | Iteration 5 | Iteration15 | Iteration 20 |
> |--------------|------------|--------------|--------------|
> |greedy| 0.38e-2 | 0.21e-2 | 0.17e-2 |
> | optimism (tour-num = 3) |  0.27e-2 | 0.14e-2 | 0.11e-2 |
>
> - On the impact of hyperparameters, the following table reports the performance comparisons under different values of $\eta$. The effectiveness of greedy sampling can be observed regardless of the value of $\eta$.
>
> | Step Regret (eta=1)   | Iteration 5 | Iteration15 | Iteration 20 |
> |--------------|------------|--------------|--------------|
> |greedy| 0.38e-2 | 0.21e-2 | 0.17e-2 |
> | optimism (tour-num = 3) |  0.27e-2 | 0.14e-2 | 0.11e-2 |
>
>
> | Step Regret (eta=2)   | Iteration 5 | Iteration15 | Iteration 20 |
> |--------------|------------|--------------|--------------|
> |greedy| 0.41e-2 | 0.31e-2 | 0.28e-2 |
> | optimism (tour-num = 3) | 0.45e-2 | 0.30e-2 | 0.26e-2 |
>
> | Step Regret (eta=3)   | Iteration 5 | Iteration15 | Iteration 20 |
> |--------------|------------|--------------|--------------|
> |greedy| 0.81e-2 | 0.46e-2 | 0.32e-2 |
> | optimism (tour-num = 3) | 0.80e-2 | 0.32e-2 | 0.30e-2 |
>
> ----
>
> **Question 1.** Zhao et al. (2025a,b) use confidence-bound constructions for BT model regret bounds. Can their approach directly yield the bounds for the GP model? If not, what modifications are essential?
>
> **Answer 3.** Thank you for this insightful comment! We believe that the optimism/pessimism-based designs for the BT model in [R1, R2] can be extended to the general preference model considered in this work, but it will not be straightforward based on the previous techniques. The analytical framework provided in this work, particularly the novel bound and decomposition of the single-step regret as discussed in Answer 1, will serve as an indispensable component for the corresponding analysis, which further highlights the contribution of our paper. We believe that the obtained results would be similar to the ones in the BT model, i.e., the performance of greedy sampling would match that of optimism/pessimism-based designs up to multiplicative constants, while greedy sampling is much more computationally favorable as it avoids the costly confidence-bound optimizations. We will add further discussions in the revised paper.
>
> ----
>
> [R1] Zhao, Heyang, et al. (2025) ``Logarithmic regret for online KL-regularized reinforcement learning.".
>
> [R2] Zhao, Qingyue, et al. (2025) ``Nearly Optimal Sample Complexity of Offline KL-Regularized Contextual Bandits under Single-Policy Concentrability."
>
> [R3] Zhao, Heyang, et al. (2024) ``Sharp analysis for kl-regularized contextual bandits and RLHF.''

---

> ### Author Response · Authors · 2025-08-07
> **Follow-up on Rebuttal Response**
>
> Dear Reviewer tKdL,
>
> We hope this message finds you well. As the rebuttal phase is nearing its end (with less than two days remaining), we would like to kindly follow up regarding our response to your comments. We would greatly appreciate it if you could let us know whether our reply has addressed your concerns adequately, or if there are any remaining questions or clarifications you'd like us to provide.
>
> Thank you again for your time and effort in reviewing our work.
>
> Best regards,
>
> Authors

---

> > ### Comment · Reviewer_tKdL · 2025-08-07
> >
> > Thank you for your detailed rebuttal.  I will maintain my positive rating.

---

> > > ### Author Response · Authors · 2025-08-07
> > >
> > > Thank you for your reply and for recognizing our efforts. We're pleased that our response clarified your concerns, and we truly appreciate your time and thoughtful comments throughout the review process.

---

### Official Review · Reviewer_dgKJ · 2025-07-04

**Clarity:** 2
**Significance:** 3
**Originality:** 3
**Rating:** 4
**Confidence:** 3

**Summary:**

This paper investigates the theoretical underpinnings of RLHF, focusing on the KL-regularized contextual bandit setting. The authors present a surprisingly strong result: a simple greedy sampling algorithm, which directly uses empirical estimates from maximum likelihood estimation without optimism or pessimism, is provably efficient. For the general preference model, the paper establishes, for the first time, a logarithmic regret bound of $O(\log(T))$ in the online setting and a sample complexity of $O(\epsilon^{-1})$ in the offline setting. The central theoretical insight is that the KL-regularization term restricts the optimal policy class, ensuring that any candidate optimal policy has a bounded likelihood ratio with respect to the reference policy $\pi_0$. This structural property mitigates the need for explicit exploration (optimism) or conservatism (pessimism).

**Questions:**

## Questions

1. Could the authors elaborate on the practical implementation of the MLE and policy update steps (*e.g.*, Step 6 and 7 in Algorithm 1) in a real-world LLM setting?
2. The analysis hinges on the optimal policy being close to the reference policy $\pi_0$. How does the quality of the initial SFT model (used as $\pi_0$) affect the final performance of the greedy algorithm? If $\pi_0$ is a poor-quality model, does that place a more severe restriction on the achievable performance for greedy sampling compared to an optimism-based approach?
3. The empirical results suggest optimism-based methods might have a slight edge in cumulative regret, likely due to better constants. In what practical scenarios would the computational simplicity of greedy sampling outweigh the potentially better constants of optimism/pessimism, and vice-versa?

**Ethical Concerns:**

["NO or VERY MINOR ethics concerns only"]

**Final Justification:**

After reading all the review comments, since I am already positive, and thus will maintain my original score.​​

**Limitations:**

See Weaknesses.

**Quality:**

3

**Strengths And Weaknesses:**

### Strengths

- The paper's core claim—that a simple greedy algorithm is provably efficient for RLHF—is both counter-intuitive and highly impactful. It challenges the prevailing wisdom that optimism or pessimism is necessary for achieving strong performance guarantees in learning from feedback. This finding could significantly simplify the design and analysis of future RLHF algorithms.
- The paper provides tight, instance-agnostic performance bounds. The $O(\log(T))$ regret and $O(\epsilon^{-1})$ sample complexity are the best possible in many learning settings. Achieving these for the general preference model for the first time is a major theoretical leap forward.
- The work is thorough in its scope, addressing both the general and BT preference models, as well as both online and offline learning settings.

### Weaknesses

- While the experiments corroborate the theoretical findings, they are conducted in a relatively simple linear setting with a small, fixed action space. The paper is motivated by LLMs, but it does not provide evidence of how this approach scales to the massive, high-dimensional, and non-linear nature of LLM fine-tuning.
- The big-O notation in the theoretical results hides constants. The authors acknowledge that the greedy sampling algorithm may have a "slightly worse constant", which is also suggested by the cumulative regret plots in Figure 1. A more direct discussion of the trade-offs between computational simplicity and these performance constants in the main body would be beneficial.
- The theoretical analysis relies on the standard but strong assumption of realizability, *i.e*., that the true preference or reward model lies within the considered function class (Assumptions 1 and 2). In practice, human preferences are incredibly complex, and model misspecification is a real concern. The paper does not discuss the robustness of the greedy approach to violations of this assumption.

---

> ### Author Rebuttal · Authors · 2025-07-31
>
> Thank you for the insightful comments. Please find our point-by-point response in the following.
>
> ----
> **Weakness 1.** While the experiments corroborate the theoretical findings, they are conducted in a relatively simple linear setting with a small, fixed action space. The paper is motivated by LLMs, but it does not provide evidence of how this approach scales to the massive, high-dimensional, and non-linear nature of LLM fine-tuning.
>
> **Answer 1.**  As the reviewer has pointed out, the main focus of this paper is to establish the theoretical properties of greedy sampling for RLHF. The empirical results are provided to corroborate the theoretical findings. As such, we consider it appropriate to focus on the relatively simple setting in the experiments, as it allows for clearer and more straightforward interpretation of empirical results. We also agree with the reviewer on the value of large-scale LLM fine-tuning experiments, which will be added to the revised paper, mainly due to the very limited rebuttal time.
>
> Also, in the following, we provide simulated results in a more complicated setting. With the general preference model considered, the number of dimensions is increased from 125 (as in Figs. 1(a) and 1(b)) to 1000 while the reward function contains random perturbations to the original linear ones. The empirical effectiveness of greedy sampling are further demonstrated through the comparisons (i.e., converging only slightly slower than the optimism-based design).
>
> | Step Regret       | Iteration 5 | Iteration 15 | Iteration 20 |
> |--------------|------------|--------------|--------------|
> | greedy | 0.38e-2 | 0.21e-2 | 0.17e-2 |
> | optimism (tour-num = 3) |  0.27e-2 | 0.14e-2 | 0.11e-2 |
>
> ----
> **Weakness 2.** The big-O notation in the theoretical results hides constants. The authors acknowledge that the greedy sampling algorithm may have a "slightly worse constant", which is also suggested by the cumulative regret plots in Figure 1. A more direct discussion of the trade-offs between computational simplicity and these performance constants in the main body would be beneficial.
>
> **Answer 2.** The complete upper bounds (including the full dependencies on $\eta$) were included in the appendix, in particular, line 968 in Appendix D.1 for Theorem 1 and line 985 in Appendix D.2 for Theorem 2. From these bounds, we can observe that greedy sampling avoids costly optimization required in optimism/pessimism-based designs while incurring an additional multiplicative constant (at most at the order of $\exp(\eta)$) in the performance bounds. Following the reviewer's suggestion, we will add these full results and corresponding discussions on the tradeoff to the main paper.
>
> ----
> **Weakness 3.** The theoretical analysis relies on the standard but strong assumption of realizability, i.e., that the true preference or reward model lies within the considered function class (Assumptions 1 and 2). In practice, human preferences are incredibly complex, and model misspecification is a real concern. The paper does not discuss the robustness of the greedy approach to violations of this assumption.
>
> **Answer 3.** Regarding the reviewer's concern, we would like to provide the following clarifications:
> - First, as the reviewer mentioned, the realizability assumption is a standard one that has been used in many theoretical RL studies, including previous theoretical investigations on RLHF (such as [R1] and [R2]). Thus, we believe it is a reasonable assumption to initiate the study on greedy sampling.
> - Second, we agree with the reviewer that in practice, human preferences are complex. This is exactly the reason that this work studied general preference model (instead of just the BT model) and general function approximation (instead of simply tabular or linear). These considerations contribute to the generality of the theoretical results in this work.
> - Furthermore, as suggested by the reviewer, it is an important direction to investigate the algorithmic robustness in RLHF when the realizability assumption is not satisfied, for both greedy sampling and previous optimism/pessimism-based designs. It can be considered that the involved function class has an error $\zeta$ to realize the preference model or the reward function; then techniques discussed in mis-specified bandits and RL (such as [R3] and [R4]) can potentially be adopted for algorithmic adaptation and analysis of the impact of the realization error $\zeta$. We will provide a more detailed discussion in the revised paper to encourage further exploration.
>
> ----
> **Question 1.** Could the authors elaborate on the practical implementation of the MLE and policy update steps (e.g., Step 6 and 7 in Algorithm 1) in a real-world LLM setting?
>
> **Answer 4.** We would like to first clarify the roles of Step 6 and Step 7 in the design. Step 6 is to obtain an estimation $\hat{P}$ of the preference model or $\hat{R}$ of the reward function through the standard maximum likelihood estimation (MLE). Then, Step 7 finds the Nash equilibrium (NE) policy associated with $\hat{P}$ under the general preference model, and the optimal policy associated with the estimation $\hat{R}$ under the BT model. Regarding the practical implementation of these two steps, some discussions are provided in the following:
>
> - For the general preference model, standard optimization techniques, such as stochastic gradient descent (SGD) and Adam can be leveraged to obtain the preference model estimation $\hat{P}$ via MLE. Furthermore, the problem of finding the NE policy associated with $\hat{P}$ has been investigated in [R5] and [R6] with practical implementations provided.
>
> - For the BT model, two standard approaches exist. First, as in the pioneering RLHF works (e.g., [R7]), the reward estimation $\hat{R}$ can be first obtained from MLE similarly through SGD or Adam, and then PPO can be adopted to optimize the policy to the reward estimation $\hat{R}$ and a KL penalty. Alternatively, DPO [R8] can be adopted to directly optimize the policy from the collected dataset.
>
> In summary, under both the general preference model and BT model, there are standard and well-developed approaches for the practical implementation of greedy sampling. This represents a significant advantage over previous optimism/pessimism-based designs, which often required costly optimizations over confidence sets and additional heuristics.
>
> ----
> **Question 2.** The analysis hinges on the optimal policy being close to the reference policy $\pi_0$. How does the quality of the initial SFT model (used as $\pi_0$) affect the final performance of the greedy algorithm? If $\pi_0$ is a poor-quality model, does that place a more severe restriction on the achievable performance for greedy sampling compared to an optimism-based approach?
>
> **Answer 5.** We would like to clarify that the targets considered in this work and previous studies (such as [R1] and [R2]) are the same, i.e., the KL-regularized target in line 161 and 178, which is the standard target in RLHF. The observation that the optimal policy is close to the reference policy is directly obtained under this same target.
>
> The differences between this work and previous ones are purely algorithmic: previous studies adopted optimism/pessimism-based algorithms, while this work leverages greedy sampling. Different algorithmic choices only impact the statistical convergence performance and computational complexities during learning, as discussed in Answer 2. Thus, under the common target, there is no restriction on the achievable performance for greedy sampling compared to optimism/pessimism-based algorithms, which we hope can alleviate the reviewer's concern.
>
> ----
> **Question 3.** The empirical results suggest optimism-based methods might have a slight edge in cumulative regret, likely due to better constants. In what practical scenarios would the computational simplicity of greedy sampling outweigh the potentially better constants of optimism/pessimism, and vice-versa?
>
> **Answer 6.** As observed from the single-step regrets reported in Figs. 1(a) and 1(c), the major performance differences between greedy sampling and the optimism-based algorithms occur at the first few iterations of training, where the single-step regrets of optimism-based designs decrease faster. However, after a few iterations (e.g., 8 iterations in Fig. 1(a) and 5 in Fig. 1(c)), both greedy sampling and optimism-based algorithms incur roughly the same single-step regrets. This observation indicates that greedy sampling is more suitable for training with limited resources but sufficient time, as it is computationally simple and achieves similar performance after the first few iterations. On the other hand, the incorporation of optimistic/pessimistic principles can be beneficial when equipped with sufficient resources but limited time. This essentially reflects a tradeoff between statistical convergence speed and computational requirements, as discussed above in Answer 2.
>
> ----
> [R1] Zhao, Heyang, et al. (2025) ``Logarithmic regret for online KL-regularized reinforcement learning.''
>
> [R2] Ye, Chenlu, et al. (2024) ``Online iterative reinforcement learning from human feedback with general preference model.''
>
> [R3] Wang, Ruosong, et al. (2020) ``Reinforcement learning with general value function approximation: Provably efficient approach via bounded eluder dimension.''
>
> [R4] Foster, Dylan J., et al. (2020) ``Adapting to misspecification in contextual bandits.''
>
> [R5] Rémi Munos, et al. (2023) ``Nash learning from human feedback.''
>
> [R6] Calandriello, Daniele, et al. (2024) ``Human alignment of large language models through online preference optimisation."
>
> [R7] Ouyang, Long, et al. (2022) ``Training language models to follow instructions with human feedback.''
>
> [R8] Rafailov, Rafael, et al. (2023) ``Direct preference optimization: Your language model is secretly a reward model.''

---

> > ### Comment · Reviewer_dgKJ · 2025-08-09
> >
> > ​Thank you to the authors for their detailed responses. After reading all the review comments, since I am already positive, and thus will maintain my original score.​​

---

> ### Author Response · Authors · 2025-08-07
> **Follow-up on Rebuttal Response**
>
> Dear Reviewer dgKJ,
>
> We hope this message finds you well. As the rebuttal phase is nearing its end (with less than two days remaining), we would like to kindly follow up regarding our response to your comments. We would greatly appreciate it if you could let us know whether our reply has addressed your concerns adequately, or if there are any remaining questions or clarifications you'd like us to provide.
>
> Thank you again for your time and effort in reviewing our work.
>
> Best regards,
>
> Authors

---

### Note · Authors · 2025-08-13

We sincerely thank the reviewers, area chair, and senior area chair for their time and thoughtful evaluations of our work. We are pleased to see that the key contributions of our paper have been clearly recognized: **all reviewers provided positive recommendations prior to the rebuttal and chose to either maintain or raise their scores afterward.**

To further facilitate the decision process, we briefly reiterate the main contributions of this work:

- **Provable Efficiency of Greedy Sampling in RLHF:** We establish that *greedy sampling based on empirical estimates is provably efficient for reinforcement learning from human feedback (RLHF)*. This result holds across *both online and offline settings*, and for *both general and Bradley-Terry (BT) preference models*. Our findings are supported by rigorous theoretical analysis and validated by experimental results.

- **Novelty and Tightness of Results:** To the best of our knowledge, this is the first time that the efficiency of greedy sampling for RLHF is proved. Furthermore, our results provide tight online regret and offline sample complexity bounds for general preference models, offering deeper theoretical insights beyond prior works.

- **Practical Implications:** The demonstrated efficiency of greedy sampling points to a promising direction for simplifying RLHF algorithms in practice. Unlike prior optimism/pessimism-based methods that rely on computing confidence bounds or solving complex optimizations, greedy sampling avoids such computation overhead, largely enhancing practical usability.

During the rebuttal phase, we provided detailed clarifications and additional supporting experimental results that addressed all raised concerns, as noted by the reviewers. These discussions and improvements will be incorporated into the revised version of the paper.

---

### Decision · Program_Chairs · 2025-09-17

**Decision:**

Accept (poster)

**Comment:**

This paper studies greedy sampling strategies for RLHF and proves tight regret bound for the online setting and sample complexity for the offline setting. The reviewers appreciated the work and mentioned that the paper shows that greedy sampling could be an effective alternative for costlier pessimism based optimization. I believe that the work could be interesting to researchers working in theoretical aspects of RL and RLHF.